# Non-Line-of-Sight 3D Reconstruction with Radar

**Haowen Lai**
University of Pennsylvania
hwlai@seas.upenn.edu

**Zitong Lan**
University of Pennsylvania
ztlan@seas.upenn.edu

**Mingmin Zhao**
University of Pennsylvania
mingminz@seas.upenn.edu

https://waves.seas.upenn.edu/projects/holoradar

## Abstract

Seeing hidden structures and objects around corners is critical for robots operating in complex, cluttered environments. Existing methods, however, are limited to detecting and tracking hidden objects rather than reconstructing the occluded full scene. We present HoloRadar, a practical system that reconstructs both line-of-sight (LOS) and non-line-of-sight (NLOS) 3D scenes using a single mmWave radar. HoloRadar uses a two-stage pipeline: the first stage generates high-resolution multi-return range images that capture both LOS and NLOS reflections, and the second stage reconstructs the physical scene by mapping mirrored observations to their true locations using a physics-guided architecture that models ray interactions. We deploy HoloRadar on a mobile robot and evaluate it across diverse real-world environments. Our evaluation results demonstrate accurate and robust reconstruction in both LOS and NLOS regions. Code, dataset and demo videos are available on the project website.

## 1 Introduction

The ability to perceive beyond direct line of sight (LOS) is essential for robots operating in complex environments. Consider an autonomous vehicle approaching an intersection as children step unexpectedly into the street. If the vehicle can perceive the hidden scene and individuals before they become visible, it can reason about their positions and motion, gaining precious time to react and prevent accidents. Similarly, indoor robots navigating tight spaces can operate more safely and efficiently by reconstructing hidden structures and obstacles around corners, thereby allowing them to plan precise, collision-free paths in advance.

Despite the importance of Non-Line-of-Sight (NLOS) perception, visual sensors such as cameras and LiDARs face fundamental limitations. These sensors operate at wavelengths on the order of hundreds of nanometers, which are much smaller than the surface roughness of common materials (e.g., drywall, concrete). As a result, light scatters in many directions, losing the directional coherence needed for imaging. In contrast, radio frequency (RF) signals have wavelengths orders of magnitude longer (i.e., millimeter or centimeter), causing most walls and structures to behave as specular reflectors like mirrors. This property enables RF signals to bounce predictably off surfaces and preserve geometric information even around corners. As illustrated in Fig. 1(a), the returning RF echoes encode rich spatial cues about hidden objects and spaces, making RF uniquely suited for NLOS perception.

Existing NLOS imaging systems face significant limitations. Radar-based approaches often require prior knowledge of the scene geometry [28, 35, 38], or rely on bulky hardware and long signal acquisition times [8, 34]. Single-photon LiDAR systems [2, 17, 24] involve expensive setups and suffer from spatial ambiguity due to diffuse reflections. Camera-based methods [3, 5, 30] analyze shadows for hidden object detection, but are highly sensitive to ambient lighting and cannot produce detailed images. More importantly, most systems [8, 28, 31, 34, 35, 38] are limited to localizing or imaging hidden objects, rather than reconstructing the full 3D geometry of the occluded scene.

In this paper, we introduce HoloRadar, a novel system that accurately reconstructs both LOS and NLOS 3D scenes with a mmWave radar. Achieving this capability requires overcoming several

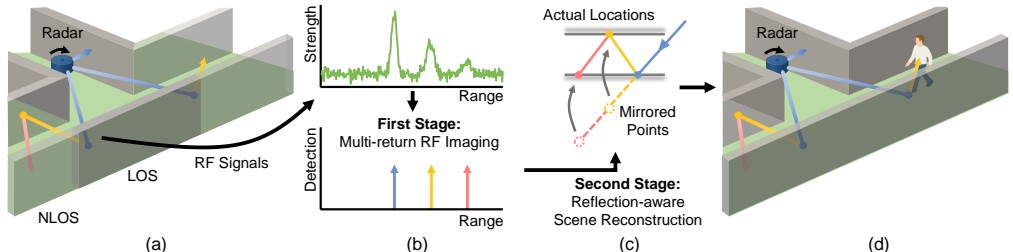

**Figure 1: HoloRadar pipeline.** It leverages (a) the multi-bounce RF reflections to reconstruct the scene. We propose a two-stage pipeline for this task. (b) The first stage predicts the range (travel distance) of each bounce from the noisy and low-resolution signals. This emulates a "multi-return LiDAR" and extracts key points in the mirrored locations. (c) The second stage maps those mirrored points to their actual locations and reconstructs the scene, (d) revealing hidden structures and humans.

technical challenges. First, radar sensors inherently suffer from low spatial resolution. Environmental reflections often appear as blurred blobs rather than distinct peaks. This is further exacerbated by measurement noise, signal artifacts, and the natural attenuation of signal strength. Second, multi-return reflections cause objects to appear at mirrored locations in the radar's measurement space, where signals from multiple bounces are superimposed in complex ways. Mapping these mirrored observations back to their true physical locations requires accurate geometry and reasoning.

Our approach addresses these challenges using a two-stage pipeline. The first stage generates high-resolution multi-return range images by predicting multiple depth values per viewing direction. For each radar beam (i.e., azimuth-elevation pair), the model estimates a set of bounce distances, each corresponding to a reflection along a multi-return path, as shown in Fig. 1(b). This is analogous to what a multi-return LiDAR system would produce if all surfaces are specular. We train this model on noisy, low-resolution radar heatmaps using supervision from ray tracing over SLAM-based scene meshes. While these multi-return range images capture geometric information from both LOS and NLOS regions, the recovered points reside in the radar's measurement space, where NLOS surfaces appear at mirrored locations due to multi-return propagation. The second stage resolves this ambiguity by reconstructing the physical 3D scene. We adopt a physics-inspired architecture that explicitly models the ray tracing process. Specifically, we combine the predicted multi-return geometry with surface normal estimation at each bounce to iteratively reverse the mirroring effect to put the mirrored points back to the world coordinates (Fig. 1(c)). This structured approach improves accuracy and accelerates convergence compared to end-to-end learning. Finally, we use a scene refinement module to correct residual geometric errors and fuse information across bounces, producing a coherent final reconstruction of the scene and hidden humans (Fig. 1(d)).

To evaluate our approach, we build a mobile robot prototype and collect data at 32 corners across 5 buildings, with 28k radar measurements in total. We train our models on 24 corners except 8 for testing. For evaluation, we measure the Chamfer distance, Hausdorff distance, and F-score between our predicted scenes and the ground truth. Our method achieves overall LOS and NLOS F-score of 85.7% and 54.6%, with 18% and 35% better over baseline methods [9, 29].

In summary, the main contributions of our paper include:

- We present the first practical system for NLOS 3D scene reconstruction around corners.
- We propose a novel two-stage pipeline that decouples signal interpretation from spatial reasoning.
- We conduct extensive experiments across diverse real-world environments and demonstrated accurate reconstruction in both LOS and NLOS regions.

## 2 Related Work

**Radar Sensing and Imaging.** Radar-based sensing and imaging have gained increasing attention in recent years due to their robustness in challenging environments [27, 39, 41]. Synthetic aperture radar (SAR) [10, 25, 26] is a technique that enhances resolution by coherently combining measurements over time or space. Several systems leverage this technique with mechanical motion. Some move the radar along horizontal and vertical axes with sliders [11, 36], while others employ rotational movement for panoramic sensing [20, 21]. For specific targets like humans [18, 22] or vehicles [11, 12, 23], machine learning models have been applied to further improve resolution. Recently, [20, 21] combine advanced signal processing with learning-based methods to support general purpose indoor imaging,

recognition, and mapping. Nevertheless, nearly all these methods work with LOS reflections, with no applicability in occluded or cluttered environments perception.

**Radar-based NLOS Perception.** Extending radar sensing to NLOS scenarios has been explored in several directions. Existing work [28, 38] localizes hidden humans by leveraging multi-bounce reflections, but it requires knowledge of the surrounding geometry. Similarly, [31] uses a LiDAR to determine the relay wall for outdoor 2D human detection and tracking. Mosaic [35], on the other hand, bypasses geometric modeling by installing retro-reflectors, yet the manual setup limits its generalizability. More recently, RFlect [8] produces coarse heatmaps of hidden objects using multi-bounce radar returns. However, it demands time-consuming mechanical scanning to simulate a large synthetic aperture. There are also through-wall sensing efforts [42, 43] using low-frequency radar, though such approaches are ineffective for around-corner scenarios where signals must traverse multiple barriers. In short, current radar-based NLOS methods are held back by deployment constraints and performance limitations. Moreover, none of these work aims to reconstruct the full 3D scene.

**NLOS Sensing with Other Modalities.** While cameras and LiDARs are widely used for LOS perception and reconstruction [19, 33, 37], recent studies have explored their potential for NLOS sensing. Several methods [3, 5, 30] use RGB cameras to analyze indirect illumination and shadows cast on visible surfaces, enabling low-resolution reconstructions of hidden human activity. However, these methods require carefully controlled lighting and wall locations, limiting their robustness in real-world conditions. Others [14–17, 24, 32] leverage single-photon avalanche diode (SPAD) LiDARs to capture two-bounce time-of-flight histograms of hidden objects through transient imaging. While these systems can provide accurate measurements, they are typically bulky and expensive for mobile or daily deployment.

## 3 Radar Imaging Background

Radar transmits electromagnetic waves and receive reflections from objects and surfaces in the environment. To achieve sufficient angular resolution, radar systems typically use antenna arrays to synthesize a large aperture. In our setup, we follow the design of PanoRadar [20], where a single-board radar is mechanically rotated to emulate a cylindrical array. As illustrated in Fig. 2(a), by compensating for signals received across antennas during a full rotation, we digitally form beams over $N_\theta$ elevation angles ($\theta$) and $N_\phi$ azimuth angles ($\phi$), where $\phi$ spans the full 360° field of view. To estimate depth along each beam, we use a frequency-modulated continuous wave (FMCW) waveform and apply a Fourier transform, resulting in $N_R$ discrete range bins. This process yields a 3D heatmap $\mathbf{H} \in \mathbb{R}^{N_\theta \times N_\phi \times N_R}$, which serves as a coarse volumetric representation of the environment. Higher values indicate stronger reflections and a greater likelihood of object presence.

Compared to camera images, radar heatmaps have significantly lower spatial resolution, and signal energy often spreads across neighboring bins [11, 23, 42]. However, radar offers a distinct sensing modality that interacts with the environment in fundamentally different ways. Due to its long wavelength, most surfaces behave as specular reflectors. As a result, signals often undergo multiple bounces before returning to the sensor, producing indirect reflections that appear at mirrored locations in the measurement space [8, 20]. While such multipath reflections are typically considered artifacts in conventional radar imaging, they carry valuable geometric information. In this paper, we build a framework that models and exploits these indirect reflections to enable accurate NLOS reconstruction.

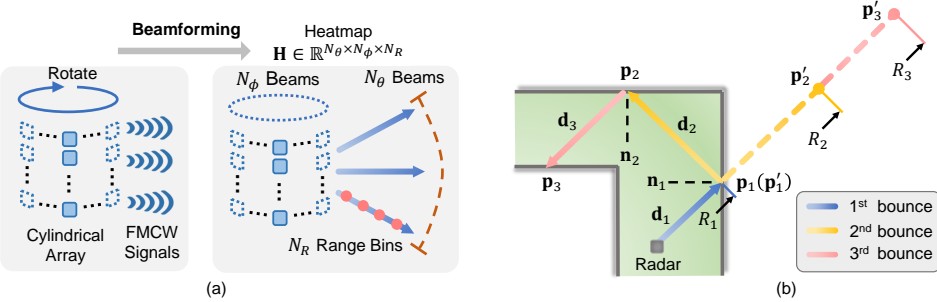

**Figure 2: System setup and multi-bounce reflection.** (a) Our rotating radar forms a cylindrical array with a 360° FOV. For signals collected in each cycle, beamforming is applied to obtain a 3D heatmap. (b) A 2D illustration of the multi-bounce reflection and notations.

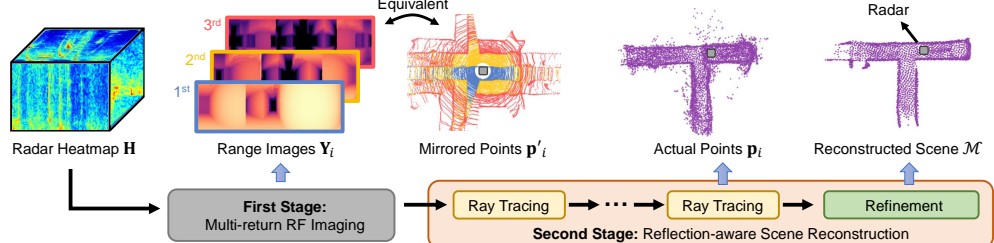

**Figure 3: Network architecture of HoloRadar.** We propose a two-stage pipeline for radar-based NLOS 3D reconstruction. The first stage aims to enhance the radar imaging resolution and emulate a "multi-return LiDAR", resulting in range images for each bounce. These images are equivalent to mirrored points. The second stage focuses on spatial reasoning, mapping mirrored points to their actual locations with a hybrid ray-tracing module, as well as refining the reconstructed scene.

# 4 Method

In this section, we first define our task and introduce notations (§ 4.1), and then show HoloRadar's two-stage decomposition network architecture (§ 4.2). This is followed by explanation of our multi-return RF imaging model (§ 4.3) and reflection-aware scene reconstruction model (§ 4.4).

## 4.1 Problem Definition and Notations

**Multi-return Reflections.** The process of how a beam reflects is illustrated in Fig. 2(b). Each beam may be reflected multiple times before returning to the radar along the same path. We use subscript $i \geq 1$ to denote the number of bounces of a beam, and the prime symbol ($'$) to indicate variables in mirror coordinate. Moreover, the beam direction for each bounce is defined as $\mathbf{d}_i$, where $\mathbf{d}_1$ is the initial beamforming direction which is known and fixed. The reflection is assumed to be perfectly specular with respect to the surface normal $\mathbf{n}_i$. We do not consider more than three bounces due to the decay of signal strength. To acquire ground truth, we use LiDAR SLAM to build a mesh map $\mathcal{M}$ for the environment.

**NLOS Imaging Task.** Our goal is to reconstruct the environment $\mathcal{M}$ from the 3D heatmap $\mathbf{H}$. Similar to an image in the mirror, the actual points $\mathbf{p}_i$ in world coordinate will appear as $\mathbf{p}'_i$ in mirror coordinate in the measurement, as shown in Fig. 2(b). While mirrored points $\mathbf{p}'_i$ lie in the same beam direction $\mathbf{d}_1$, the actual points $\mathbf{p}_i$ can be anywhere depending on the environmental structures, radar's location, and the number of bounces. This task is further complicated by the low-resolution noisy radar heatmaps as inputs.

## 4.2 HoloRadar Architecture

To tackle this problem, HoloRadar decomposes it into two sub-tasks, and adopts a two-stage structure to handle each one accordingly, as shown in Fig. 3. The **first stage** is multi-return RF imaging, which is responsible for identifying the mirrored points $\mathbf{p}'_i$ from the heatmap that is low-resolution with plenty of noise and artifacts. This can be formulated as a range estimation task, as $\mathbf{p}'_i = R_i \mathbf{d}_1$, where $R_i$ is the accumulated range (or travel distance) of a beam up to the $i$-th bounce. If we estimate the accumulated ranges for every initial direction $(\mathbf{d}_1)_{\theta,\phi}$, it will form three dense range images $\mathbf{Y}_i = (R_i)_{\theta,\phi} \in \mathbb{R}^{N_\theta \times N_\phi}$. Each pixel in $\mathbf{Y}_i$ is essentially a mirrored point $\mathbf{p}'_i$. Fig. 3 visualizes an input heatmap and the predicted range images that correspond to mirrored point clouds.

The **second stage** is reflection-aware scene reconstruction, which is designed to reason about the spatial relationship between the real world and the mirror world and to put mirrored points $\mathbf{p}'_i$ back to their actual locations $\mathbf{p}_i$. To achieve this, we take inspiration from the physical reflection rules and introduce the ray tracing blocks with physical inductive bias. These modules work in a cascaded manner, where the current ray tracing block takes the predicted reflected directions and surface normals from the previous block and outputs them for the next one. Hence, with the travel distance (accumulated range $R_i$) of each beam from the first stage, the actual locations $\mathbf{p}_i$ can be determined sequentially for every bounce. After this stage, points are placed back to the world coordinate. However, there are still some noisy and misaligned points due to the sequential prediction and error accumulations. We further apply a scene refinement module to denoise the results and fuse points from all bounces for final scene reconstruction.

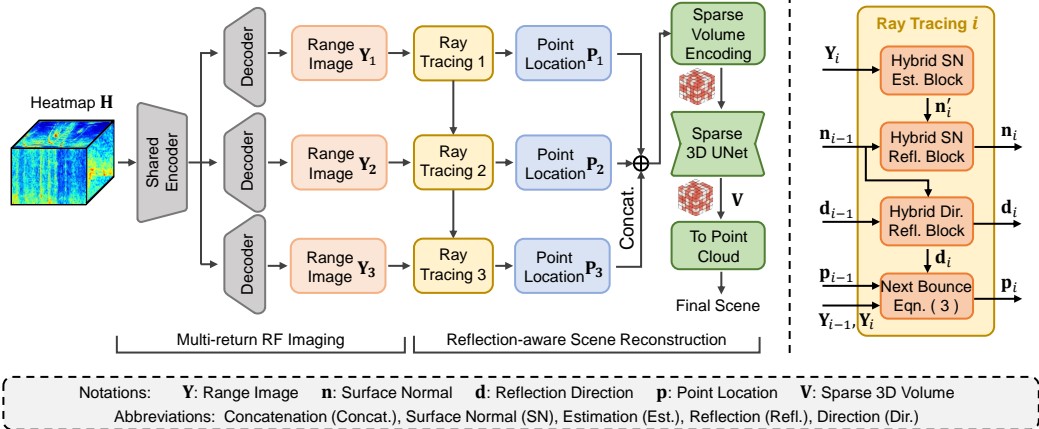

**Figure 4: Network Details.** *Left:* HoloRadar adopts a two-stage pipeline, taking an RF heatmap as input and predicting the scene around corners. *Right:* Details of the $i$-th ray tracing block and its inputs and outputs.

### 4.3 Multi-return RF Imaging

The multi-return RF imaging model focuses on dense range estimation, aiming to predict the range images $\mathbf{Y}_i \in \mathbb{R}^{N_\theta \times N_\phi}$, $i = 1, 2, 3$ from the 3D radar heatmap $\mathbf{H} \in \mathbb{R}^{N_\theta \times N_\phi \times N_R}$. To process the heatmap, we adopt the strategy of treating the range dimension as channels and apply 2D CNNs across elevation and azimuth dimension, an approach shown to be effective for enhancing imaging resolution in prior work [20]. This design is well-suited to address the spatial energy spreading caused by the limited sensing resolution while remaining computationally efficient.

Building on this, we develop a UNet-based architecture with a shared encoder, separate decoders, and skip connections, as illustrated in Fig. 4. The encoder processes the full heatmap $\mathbf{H}$ and extracts shared RF features, which are then passed through three specialized decoders to reconstruct the respective range images $\mathbf{Y}_i$ for each bounce. Our insight is that while LOS and NLOS regions often share signal patterns caused by similar environmental structure, they still differ in several aspects such as strength, noise characteristics, and energy spreading. Thus, instead of regressing a multi-channel image directly, we design separate decoders to better capture the distinct signal characteristics of different bounces. With the predicted range images $\hat{\mathbf{Y}}_i$, we compute the loss as a combination of per-pixel range estimation error and perceptual error:

$$\mathcal{L}_{\text{img}} = \lambda_{\text{range}} \sum_i \left\| \hat{\mathbf{Y}}_i - \mathbf{Y}_i \right\|_1 + \lambda_{\text{lpips}} \sum_i \texttt{LPIPS}(\hat{\mathbf{Y}}_i, \mathbf{Y}_i), \tag{1}$$

where $\lambda_{\text{range}}$, $\lambda_{\text{lpips}}$ are weighting factors, and $\texttt{LPIPS}(\cdot, \cdot)$ computes the perceptual loss using LPIPS metrics [40]. The three branches are jointly optimized, and the perceptual loss is used for capturing details objects such as humans. We obtain the ground truth $\mathbf{Y}_i$ by performing ray tracing on the environment mesh map $\mathcal{M}$, with initial directions the same as the beamforming directions $\mathbf{d}_1$.

### 4.4 Reflection-aware Scene Reconstruction

After obtaining the range images $\mathbf{Y}_i = (R_i)_{\theta, \phi}$, the reflection-aware scene reconstruction model first maps every mirrored points $\mathbf{p}'_i = R_i \mathbf{d}_1$ back to their actual location $\mathbf{p}_i$ using our ray tracing modules. These modules work in a cascaded manner and sequentially predict the actual points. Then, it refines all $\mathbf{p}_i$ to reconstruct the final scene in volumetric representation, and converts it to point cloud for visualization. Fig. 4 shows the detailed structure of the model.

**Ray Tracing.** This module simulates how a ray bounces off a surface and change its direction, assuming perfect specular reflection. Given the previous incoming ray direction $\mathbf{d}_{i-1}$ and the surface normal $\mathbf{n}_{i-1}$ at point $\mathbf{p}_{i-1}$, the current outgoing reflection direction $\mathbf{d}_i$ can be determined by:

$$\mathbf{d}_i = \mathbf{d}_{i-1} - 2(\mathbf{d}_{i-1} \cdot \mathbf{n}_{i-1})\mathbf{n}_{i-1}. \tag{2}$$

The reflected ray then travels along the current direction $\mathbf{d}_i$ for distance $R_i - R_{i-1}$ until it hits a surface, giving the current hit point $\mathbf{p}_i$ as:

$$\mathbf{p}_i = \mathbf{p}_{i-1} + (R_i - R_{i-1})\mathbf{d}_i. \tag{3}$$

Equations (2) and (3) show that the mirrored points $\mathbf{p}'_i = R_i \mathbf{d}_1$ from multiple bounces can be iteratively mapped to their actual location. Therefore, we duplicate the ray tracing module three times and cascade them for the prediction of all the points $\mathbf{p}_i$ in world coordinate.

During ray tracing, the surface normal $\mathbf{n}_i$ is unknown and needs estimation. For a range image $\mathbf{Y}_i = (R_i)_{\theta,\phi}$, the surface normal at direction $(\theta, \phi)$ has a close-form solution [1] as:

$$\mathbf{n}'_i = \mathbf{R}_{\theta,\phi} \mathbf{b}_i, \quad \mathbf{b}_i = \left[ 1, \frac{1}{R_i \cos\theta} \frac{\partial R_i}{\partial \phi}, \frac{1}{R_i} \frac{\partial R_i}{\partial \theta} \right]^{\mathrm{T}}, \tag{4}$$

where $\mathbf{R}_{\theta,\phi}$ is a pre-computed rotation matrix for every direction $(\theta, \phi)$ and $\mathbf{b}_i$ is a geometry vector depending on the local surface structure. Note that the normal vector $\mathbf{n}'_i$ in Eqn. (4) is a mirrored version of the true surface normal $\mathbf{n}_i$, because $\mathbf{Y}_i$ represents an image of the mirrored world. To recover the actual normal $\mathbf{n}_i$ in the world coordinate, we apply a reflection operation using the previously computed normal $\mathbf{n}_{i-1}$:

$$\mathbf{n}_i = \mathbf{n}'_i - 2(\mathbf{n}'_i \cdot \mathbf{n}_{i-1})\mathbf{n}_{i-1}. \tag{5}$$

For implementation purposes, we manually initialize $\mathbf{Y}_0 = \mathbf{n}_0 = \mathbf{p}_0 = \mathbf{0}$, and set $\mathbf{d}_0 = \mathbf{d}_1$, which is the known and fixed beamforming directions.

**Hybrid Estimation Blocks.** Since surface normal calculation in Eqn. (4) involves derivative, it is sensitive to noise and errors in the range images. However, accurate surface normal is critical for determining the next bounce point. That is because an error in surface normal will result in incorrect reflection directions, and this error will be further magnified for longer distance. To address this issue, we add a residual neural network branch for Eqn. (2)(4)(5), extracting features directly from the input. These input are similar to images but with different channel dimensions, representing the surface normal directions ($\mathbf{n}_{i-1}$) and range ($\mathbf{Y}_i$). The extracted features and analytical results are concatenated and fused together with fusion blocks. Both the residual branch and the fusion branch are made of standard residual blocks [13]. Structure details can be found in Appendix § C.2. Ablation study in § 5.4 demonstrate the effectiveness of our hybrid learning design.

**Refinement and Scene Reconstruction.** After ray tracing, the predicted points $\mathbf{p}_i$ are now in world coordinates. However, some of them may not align precisely with the geometry. To address this, we incorporate a voxel-based 3D UNet that fuses predictions from different bounces and refine the reconstruction with volumetric representation. Given that most voxels are empty ($\sim$90%), we adopt sparse 3D convolutions from Minkowski Engine [7] to reduce computation cost and memory usage.

We formulate scene reconstruction as a hybrid multi-class classification and regression task. Each input point is associated with a vector $(x,y,z,\Delta x,\Delta y,\Delta z,i,n)$, where $(x,y,z)$ are the point coordinates, $(\Delta x,\Delta y,\Delta z)$ are offsets to the voxel center, $i$ is the bounce index, and $n$ is the number of points falling into the voxel. A voxel encoder projects this value into feature vector, which is then averaged to obtain the voxel feature when multiple points fall into the same voxel. Bounce index $i$ is included to help the model assess the reliability of each point, since different bounce point has various noise distribution. The model predicts a classification volume $\mathbf{V}_{\mathrm{cls}}$ and a regression volume $\mathbf{V}_{\mathrm{reg}}$. We distinguish three semantic classes: empty space, humans, and structures. The final 3D positions of all non-empty voxels are obtained by adding the regressed offset to their corresponding voxel centers.

**Model Training.** We supervise both the ray tracing and the refinement module to improve performance and accelerate convergence. In the ray tracing module, instead of directly supervising the predicted surface normal $\hat{\mathbf{n}}_i$, we compute a loss on the corresponding geometry vector $\hat{\mathbf{b}}_i$ against the ground truth $\mathbf{b}_i$. We also use a per-point L1 loss between the predicted points $\hat{\mathbf{p}}_i$ and the ground truth points $\mathbf{p}_i$. The total loss for ray tracing is defined as:

$$\mathcal{L}_{\mathrm{RT}} = \lambda_{\mathrm{sn}} \sum_i \|\hat{\mathbf{b}}_i - \mathbf{b}_i\|_1 + \lambda_{\mathrm{point}} \sum_i \|\hat{\mathbf{p}}_i - \mathbf{p}_i\|_1. \tag{6}$$

For scene refinement module, we apply a cross-entropy loss to the classification prediction $\hat{\mathbf{V}}_{\mathrm{cls}}$ and an L1 loss to the offset regression prediction $\hat{\mathbf{V}}_{\mathrm{reg}}$. To encourage the predicted non-empty voxels to align closely with surfaces, we introduce a distance loss. Specifically, we apply a Euclidean Distance Transform (EDT) [4] to the ground truth volume $\mathbf{V}_{\mathrm{cls}}$, which assigns each empty voxel with distance to the nearest non-empty voxel. The total loss for refinement becomes:

$$\mathcal{L}_{\mathrm{refine}} = \lambda_{\mathrm{cls}} \mathrm{CE}(\hat{\mathbf{V}}_{\mathrm{cls}}, \mathbf{V}_{\mathrm{cls}}) + \lambda_{\mathrm{reg}} \|\hat{\mathbf{V}}_{\mathrm{reg}} - \mathbf{V}_{\mathrm{reg}}\|_1 + \lambda_{\mathrm{dist}} \|\hat{\mathbf{V}}_{\mathrm{cls}} * \mathrm{EDT}(\mathbf{V}_{\mathrm{cls}})\|_1, \tag{7}$$

where $\mathbf{V}_{\mathrm{cls}}$ and $\mathbf{V}_{\mathrm{reg}}$ denote the ground truth classification and regression volumes, respectively, and the symbol $*$ denotes element-wise (per-voxel) multiplication.

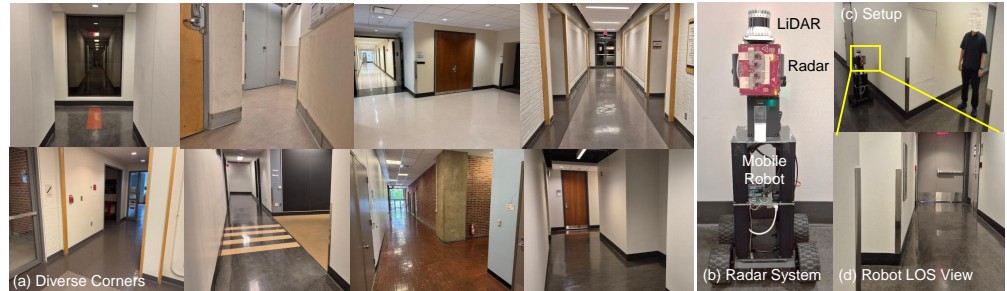

**Figure 5: Experiment setup.** (a) Example corners in the dataset. (b) Our radar system and robot. (c) Around-corner setup. (d) The human is occluded from the robot's LOS view.

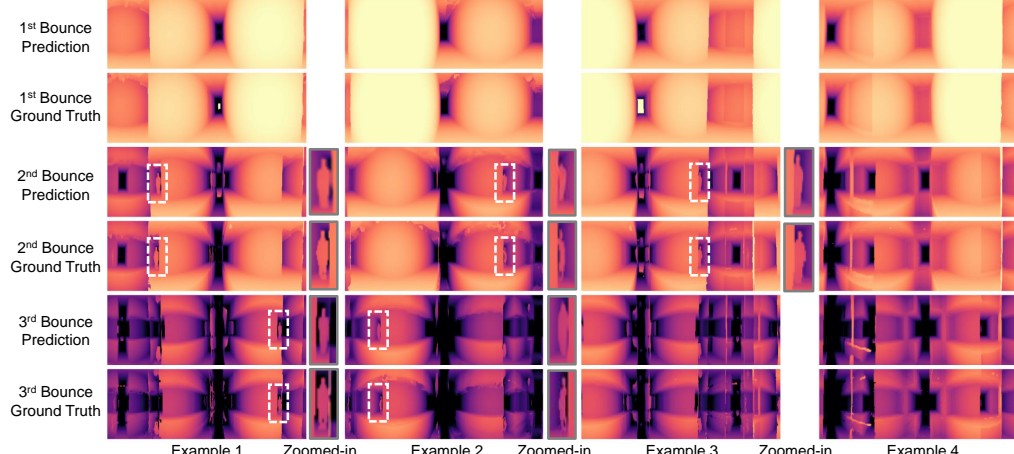

**Figure 6: Multi-return RF imaging results.** Our model successfully reveals NLOS structures and human (highlighted and zoomed in by white boxes) in both the $2^{nd}$ and $3^{rd}$ bounce images.

## 5 Experiments

### 5.1 Experimental Setup

**Dataset.** We collect a dataset from 32 distinct corners across 5 buildings, which are constructed between 1906 and 1996 and renovated between 1973 and 2017. We use 24 corners for model training and the remaining 8 for evaluation. As illustrated in Fig. 5(a), our dataset includes diverse corner layouts, including 21 T-shaped, 5 L-shaped, 5 cross-shaped, and 1 oblique corner at 45°. Corner width ranges from 1.33 m to 4.63 m, with a mean of 2.16 m and a standard deviation of 0.89 m. In each corner, we positioned a human subject behind the corner, out of the direct LOS, to simulate realistic NLOS imaging conditions. Fig. 5(b) shows our radar system, and examples of dataset collection scenarios are depicted in Fig. 5(c) and (d). Both the human and the robot are free to move, leading to a total of 28k distinct RF heatmap scans. To capture humans located in regions invisible to LiDAR, we recorded videos using a stereo depth camera, and then aligned the camera point cloud with the environmental mesh using point cloud registration [6].

**Evaluation Metrics and Baselines.** We employ comprehensive depth error metrics, including mean (cm), median (cm), 90-percentile error (cm), and PSNR (dB), to evaluate the performance of the first stage, multi-return RF imaging. In the second stage of scene reconstruction, we assess the results using Chamfer Distance (CD, cm), Modified Hausdorff Distance (MHD, cm), and F-score (F-S., %), which quantifies geometric agreement between predicted and ground truth point clouds. We note that existing radar-based see-around-corner methods neither predict dense multi-return images nor reconstruct the full scene, therefore unsuitable for direct comparison. To provide fair baselines, we implemented two vision transformer models, ViT [9] and DPT [29]. Details on the evaluation metrics (§ B.4) and the baseline implementation (§ C.3) can be found in the appendix.

**Implementation Details.** We individually train the models in the two stages. For multi-return RF imaging, both encoder and decoder architectures consist of 7 layers, with each encoder layer containing 4 residual blocks and each decoder layer containing 2 residual blocks. We trained this model using a batch size of 8 for 90k iterations. In reflection-aware scene reconstruction, our hybrid

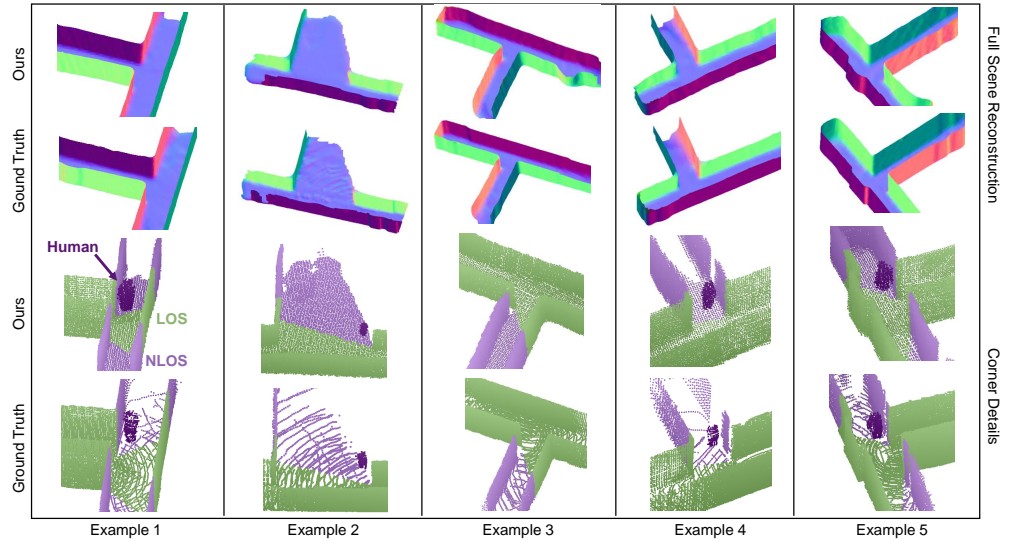

**Figure 7: Scene reconstruction results.** Top two rows: full scene reconstruction. Bottom two rows: detailed views of each corner. Green and purple points represent LOS and NLOS geometries, respectively, while dark-purple points indicate the hidden person.

estimation blocks incorporate 4 residual blocks each in both the residual and fusion branches. Our refinement model adopts a 5-layer 3D UNet structure, with each layer composed of 2 residual blocks using sparse convolution in both encoder and decoder stages. The scene space is set as $20 \, \text{m} \times 20 \, \text{m} \times 3.5 \, \text{m}$ and partitioned into $256 \times 256 \times 32$ voxels. We train this model with a batch size of 4 for 60k iterations while freezing the first stage. All experiments use the AdamW optimizer, incorporating a warm-up period of 1k steps and an initial learning rate of $10^{-4}$ following a cosine annealing schedule. All models are trained on an NVIDIA L40 GPU.

## 5.2 Results

**Multi-return RF Imaging.** We present the results of multi-return RF imaging in Tab. 1, comparing range estimation errors for each bounce individually. The results indicate that our UNet-based multi-return RF imaging model achieves superior range estimation accuracy across all three bounces. As shown in Fig. 6, the predicted range images align closely with the ground truth. Notably, in both the second- and third-bounce cases, our model accurately reconstructs depth maps, capturing the hidden human silhouettes (in white boxes) that closely match the ground truth.

| Method | 1st Bounce | | | | 2nd Bounce | | | | 3rd Bounce | | | |
|--------|-------|---------|-------|-------|-------|---------|-------|-------|-------|---------|-------|-------|
|  | Mean↓ | Median↓ | 90th↓ | PSNR↑ | Mean↓ | Median↓ | 90th↓ | PSNR↑ | Mean↓ | Median↓ | 90th↓ | PSNR↑ |
| ViT | 9.81 | 2.69 | 19.83 | 31.37 | 24.30 | 9.44 | 57.15 | 25.03 | 40.61 | 17.68 | 98.08 | 22.11 |
| DPT | 8.87 | 2.37 | 16.68 | 31.44 | 23.00 | 8.45 | 51.48 | 24.88 | 39.27 | 15.94 | 87.09 | 21.64 |
| Ours | **7.03** | **2.02** | **12.33** | **32.83** | **19.03** | **7.17** | **40.05** | **25.98** | **31.36** | **13.72** | **73.95** | **22.58** |

**Table 1:** Performance of the multi-return RF imaging model.

**3D Scene Reconstruction.** After training the first stage, we ]train the reflection-aware scene reconstruction module and evaluate its performance. As shown in Tab. 2, our two-stage pipeline consistently outperforms end-to-end transformer baselines across all metrics, in both LOS and NLOS regions. These results validate that: (i) decomposing the task into distinct imaging and spatial reasoning stages and (ii) incorporating a physics-guided ray-tracing bias significantly enhance model generalization. Even prior to the scene refinement stage, our hybrid ray-tracing module already surpasses the best baseline. The subsequent scene refinement further aligns the reconstructed point cloud with the ground truth, improving the LOS and NLOS F-scores by approximately 18% and 35% over competing methods, respectively. Fig. 7 presents qualitative results of reconstructed scene geometries from various test corners. The predicted reconstructions align closely with the ground truth, accurately capturing both hidden humans and surrounding NLOS structures. This demonstrates that our method effectively extends perception from direct LOS to occluded NLOS regions. Comparisons with baseline methods are shown in Fig. 8, where our approach produces

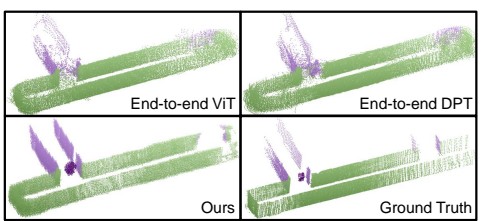

Figure 8: Visual comparison with baselines.

| Method | LOS | | | NLOS | | |
|---|---|---|---|---|---|---|
| | CD↓ | MHD↓ | F-S.↑ | CD↓ | MHD↓ | F-S.↑ |
| End2end ViT | 19.8 | 13.1 | 72.4 | 45.4 | 28.5 | 40.4 |
| End2end DPT | 21.5 | 14.6 | 69.8 | 50.3 | 31.1 | 37.0 |
| w/o Refine. | 13.8 | 7.3 | 83.8 | 43.5 | 29.3 | 50.3 |
| w/ Refine. | **12.2** | **7.1** | **85.7** | **40.0** | **28.3** | **54.6** |

Table 2: Results for the final 3D scene reconstruction.

clearer and more complete reconstructions in both LOS and NLOS areas, while the baselines exhibit noisy or incorrect structures (e.g., missing humans or closed corridors).

**Human Detection and Localization.** Our system predicts human labels on the reconstructed point cloud, enabling hidden human detection and localization. Detection performance is evaluated using a 50 cm matching threshold, where detected human points are aggregated into centroids and compared to ground-truth positions using Euclidean distance. Our method achieves a detection precision of 89.96%, recall of 97.29%, and F-score of 93.48%, with a mean localization error of 13.77 cm and a median of 12.55 cm. The high detection rate and low position error demonstrate strong performance in NLOS human detection and localization, allowing robots to navigate safely around corners.

### 5.3 Analysis on Corner Diversity and Signal Attenuation

To evaluate the robustness and generalization of our system, we test NLOS reconstruction performance across corners with varying shapes, sizes, and signal attenuation levels. The results are summarized in Tab. 3.

**Corner Shapes and Sizes.** Corners are categorized by shape (T-shaped, cross-shaped, and L-shaped) and by size (narrow: <1.5 m, medium: 1.5–3 m, wide: >3 m). We observe that T-shaped corners yield the best performance, likely due to their larger reflective surfaces, which facilitate stronger multi-bounce signal propagation. Cross-shaped and L-shaped corners, with more irregular geometries, show slightly reduced accuracy. Among the size categories, medium-width corners achieve the lowest reconstruction errors. This is because narrow corners may

| | Variation | CD↓ | MHD↓ | F-S.↑ |
|---|---|---|---|---|
| Shape | T-shaped | **34.1** | **23.9** | **61.9** |
| | Cross-shaped | 37.8 | 24.9 | 47.8 |
| | L-shaped | 45.3 | 32.7 | 52.5 |
| Size | Narrow | 39.1 | 27.5 | 57.6 |
| | Medium | **33.0** | **21.7** | 57.5 |
| | Wide | 42.5 | 32.8 | **59.2** |
| Signal Atten. | High | 39.4 | 27.1 | 53.1 |
| | Medium | 37.2 | 26.4 | 55.9 |
| | Low | **35.1** | **25.5** | **66.1** |

Table 3: NLOS reconstruction performance by different corner shapes, sizes, and signal attenuation levels.

introduce stronger multipath interference, while wide corners can weaken reflections due to longer propagation paths and greater beam dispersion. Nevertheless, the observed variation remains modest, approximately 10 cm in Chamfer and Hausdorff distances, and around 10% in F-score, indicating that our system maintains accurate even under structural irregularities.

**Signal Attenuation Levels.** Different materials exhibit varying levels of RF signal attenuation. To quantify attenuation, we sample the radar heatmap in the LOS region and compare it with the energy observed in the hidden target region, measuring the power drop between LOS and corresponding NLOS reflections. Based on this measure, the test data are divided into three bands: low attenuation (<10 dB), medium attenuation (10–20 dB), and high attenuation (>20 dB). As expected, the system performs best in low-attenuation conditions. However, even under substantial signal loss, it maintains low Chamfer and Hausdorff distances and a high F-score. These results demonstrate the resilience of our method to challenging reflective conditions, and the robustness in diverse environments.

### 5.4 Ablation Study

**Model Architecture for Multi-return RF Imaging.** We study the impact of different encoder and decoder layers on multi-return RF imaging performance, as summarized in Tab. 4. Each configuration is defined by the number of shared encoder layers, separate decoder layers, and shared decoder layers. The total number of decoder layers (separate plus shared) is kept equal to the number of encoder layers for consistency. Deviating from our chosen configuration, by either reducing or increasing the layer count, leads to diminished or saturated performance. The last three rows of the table further analyze the effect of increasing bounce-specific decoder layers while keeping the encoder depth fixed. Results show that adding separate decoder layers for each bounce consistently improves accuracy across all bounces, supporting our design choice discussed in § 4.3.

| Settings | 1st Bounce | | | | 2nd Bounce | | | | 3rd Bounce | | | |
|---|---|---|---|---|---|---|---|---|---|---|---|---|
| | Mean↓ | Median↓ | 90th↓ | PSNR↑ | Mean↓ | Median↓ | 90th↓ | PSNR↑ | Mean↓ | Median↓ | 90th↓ | PSNR↑ |
| (4,4,0) | 8.15 | 2.17 | 14.74 | 31.58 | 22.02 | 7.68 | 47.37 | 24.76 | 35.95 | 14.76 | 86.61 | 21.47 |
| (7,7,0) | **7.03** | **2.02** | **12.33** | **32.83** | **19.03** | 7.17 | **40.05** | **25.98** | 31.36 | 13.72 | **73.95** | 22.58 |
| (10,10,0) | 7.14 | 2.06 | 12.67 | 32.72 | 19.11 | **7.05** | 40.30 | 25.86 | **31.28** | **13.51** | 74.18 | **22.62** |
| (7,0,7) | 7.61 | 2.23 | 13.51 | 32.33 | 19.41 | 7.27 | 41.34 | 25.92 | 31.64 | 13.84 | 76.10 | 22.54 |
| (7,3,4) | 7.25 | 2.09 | 12.88 | 32.67 | 19.07 | **7.09** | 40.24 | 25.86 | 31.40 | 13.78 | 74.34 | 22.55 |
| (7,7,0) | **7.03** | **2.02** | **12.33** | **32.83** | **19.03** | 7.17 | **40.05** | **25.98** | 31.36 | 13.72 | **73.95** | 22.58 |

**Table 4: Ablations for multi-return RF imaging.** We evaluate different configurations of (#encoder layers, #separate decoder layers, #shared decoder layers). Colored lines are our default configuration.

**Module Design for Ray Tracing.** As shown in Tab. 5, we evaluate various designs for our hybrid ray-tracing module, including pure signal processing (SP only), pure neural networks (ML only), and our proposed hybrid approach. Our hybrid method significantly outperforms either the pure signal processing or pure neural network approaches, validating the discussion presented in § 4.4.

| Study objectives | Variation | NLOS F-S.↑ |
|---|---|---|
| Ray tracing | SP only | 16.2 |
| | ML only | 40.4 |
| Scene refinement | w/o distance loss | 29.3 |
| | w/o offsets | 50.7 |
| Our implementation | | **54.6** |

**Table 5:** Ablations for the reflection-aware scene reconstruction model.

**Design of Scene Refinement Module.** We further analyze the scene refinement module by varying the loss function and network output targets, as detailed in Tab. 5. Introducing a distance-based loss significantly improves the F-score, demonstrating that this loss encourages predicted voxel locations to more accurately align with ground truth. Additionally, predicting continuous voxel coordinates rather than discrete quantized locations helps mitigate errors due to spatial quantization.

**3D Convolution Efficiency.** To assess our choice of sparse 3D convolution, we evaluate the trade-off between reconstruction accuracy and computational efficiency across different convolution designs. We compare dense 3D convolutions at multiple feature resolutions with our sparse convolution model. All inferences are performed on an NVIDIA RTX 4070 GPU, and results are summarized in Tab. 6. The sparse 3D convolution achieves substantially faster inference than full-resolution dense convolution while maintaining comparable accuracy. Although it requires an additional rulebook, the inherent sparsity of the scene significantly reduces computation time, validating this design choice. The end-to-end inference time is 298 ms (38.4 ms for multi-return imaging and 80 ms for ray tracing), which is within the radar's 0.5 s rotation period and supports real-time operation.

| Design choices | Infer. time (ms) | GFLOPS | #Parameters | NLOS CD↓ | NLOS F-S.↑ |
|---|---|---|---|---|---|
| Full-resolution dense 3D conv | 274 | 622.24 | 54.7 M | **37.5** | 53.7 |
| 2x downsampled 3D conv | 121 | 322.33 | 54.7 M | 37.6 | 47.1 |
| 4x downsampled 3D conv | **40** | 247.61 | 54.7 M | 42.7 | 29.9 |
| Sparse 3D conv (ours) | 180 | **225.05** | **53.1 M** | 40.0 | **54.6** |

**Table 6:** Runtime and NLOS reconstruction performance for 3D convolution design choices.

# 6 Discussions

**Conclusion.** We introduced HoloRadar, a radar-based system capable of reconstructing both LOS and NLOS 3D scenes around corners. Leveraging a two-stage learning and physics-guided framework that decouples signal interpretation from spatial reasoning, HoloRadar accurately reconstructs hidden geometry and human subjects with high fidelity. Extensive evaluations on a real-world dataset covering diverse corner scenarios validate the effectiveness and robustness of the proposed approach. By enabling full-scene reconstruction beyond direct visibility, HoloRadar advances radar sensing toward broader spatial perception. This capability supports safer autonomous navigation, improved situational awareness in search and rescue, and more effective human-robot interaction in complex or visually occluded environments.

**Limitations.** As an initial step toward NLOS 3D scene reconstruction, this work focuses primarily on indoor around-corner scenarios. Extending the system outdoors would introduce challenges such as longer signal propagation distances and interference from fast dynamic objects. However, the proposed two-stage pipeline can be readily adapted to radar systems designed for outdoor operation. In addition, the current scene refinement module lacks an explicit human-shape prior. Integrating parametric human body models could further enhance the fidelity of reconstructed human geometry.

## Acknowledgments and Disclosure of Funding

We sincerely thank the reviewers for their insightful comments and suggestions. We are grateful to Zhiwei Zheng for the discussion, and to Running Zhao and Xin Yang for their help in data collection. This work is supported by University of Pennsylvania with no competing interest.

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

# A   Additional Experimental Results

We provide further experimental comparisons between baseline methods and our approach. Fig. 10 illustrates the multi-return RF imaging results for each bounce. Our predictions align closely with the ground truth, whereas baseline methods exhibit several noticeable artifacts, marked by gray boxes. Specifically, examples 1 and 4 highlight incorrect geometric predictions, example 2 demonstrates a missed detection of a human subject, and example 3 indicates an additional, incorrect detection. These inaccuracies can negatively impact subsequent scene reconstruction tasks. The comparison clearly shows the robustness and effectiveness of our design.

Additional scene reconstruction results are showcased in Fig. 11. Baseline methods achieve rough geometry reconstructions but generate significant noise, particularly at greater distances. Additionally, these methods consistently fail to detect humans located around corners, limiting their real-world applicability for robotic tasks. In contrast, our method produces clean and accurate reconstructions of both LOS and NLOS scenes, and reliably detects humans in NLOS regions. This outcome underscores the effectiveness of our proposed two-stage approach compared to end-to-end methods.

# B   System Implementation Details

## B.1   Radar Configuration

Our system employs a millimeter-wave (mmWave) cascaded radar, specifically the TI MMWCAS-RF-EVM combined with a TI MMWCAS-DSP-EVM data capture board, to transmit radio signals and receive environmental reflections. The radar frequency sweeps from 77 to 81 GHz, resulting in a 4 GHz bandwidth and a range resolution of 3.75 cm. Each chirp consists of 256 samples, allowing a maximum sensing range of 9.6 m.

The radar is mechanically rotated by a stepper motor at a frequency of 2 Hz with a rotation radius of 10.5 cm. For the raw radar measurements collected over a full cycle, they will be processed into an RF heatmap, which serves as the input to the neural network. During signal processing, beamforming is conducted over 64 elevation angles spanning [-45°, 45°] and 1024 azimuth angles covering 360°. The data is further downsampled by 2x along the azimuth dimension, resulting in a final 3D heatmap $\mathbf{H}$ of dimensions 64×512×256 (elevation, azimuth, and range respectively). The heatmap retains only amplitude data and omits phase information, since amplitude directly indicates reflection strength. An example heatmap is shown in Fig. 9(a), where we select a 2D range-azimuth slice at zero elevation.

## B.2   Data Collection

To establish ground truth, our setup includes an Ouster 64-beam LiDAR sensor alongside the mmWave radar. Both sensors are mounted on a Lynxmotion Wheeled Rover mobile robot platform, which is manually operated using a joystick.

During data collection, a human subject stands on one side of a corner while the robot positions itself on the opposite side, with both entities free to move. As the human is obscured by the corner in NLOS, we use a ZED 2i stereo depth camera to record visual data of the subject. The stereo videos are further processed to give camera point clouds. Radar, LiDAR, and camera data are collected simultaneously and synchronized using timestamps. Our dataset comprises data from 32 different

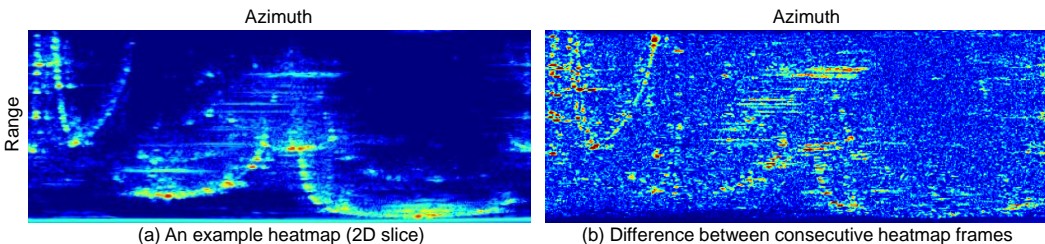

(a) An example heatmap (2D slice)          (b) Difference between consecutive heatmap frames

**Figure 9: Visualization of RF heatmaps.** (a) A 2D range-azimuth slice at zero elevation of an example RF heatmap. (b) The absolute amplitude difference between consecutive heatmap frames.

corners, each containing roughly 875 synchronized data collection cycles, totaling approximately 28k samples. To simulate real-world variability, some data are collected without human presence.

To capture variation across scans of the same corner, data are collected with both a moving robot and moving human, causing sensor and target positions to vary across acquisitions. At millimeter-wave frequencies, even millimeter-scale position changes can shift the signal phase, leading to noticeable differences in the beamformed RF heatmaps between scans. Fig. 9(b) illustrates these effects, showing the absolute amplitude difference between consecutive frames.

### B.3 Ground Truth Acquisition

Ground truth data for LOS and NLOS scene reconstruction are generated using a customized ray tracing pipeline. Initially, we leverage a 3D LiDAR SLAM algorithm to generate a detailed geometric mesh map of each corner without human presence. Then, we register point clouds from the depth camera to the pre-built mesh map to determine the initial camera pose. This is a one-time calibration since the camera is stationary. LiDAR point clouds are similarly registered, but it is performed dynamically for each rotation cycle due to robot movements. This ensures consistent spatial alignment of both robot and human subjects within the shared map coordinate system.

We then simulate beamforming directions through ray tracing. For each ray cast, we record travel distances at every bounce, generating accurate ground truth multi-return range images. Additionally, surface normals, semantic labels (LOS, NLOS, human, non-human), and spatial coordinates (x, y, z) of each intersection point are computed and stored.

### B.4 Evaluation Metric

For reconstruction results, we evaluate the similarity between a predicted point cloud $\mathbf{P}$ and its ground-truth counterpart $\mathbf{G}$ using three metrics: the Chamfer Distance (CD), the Modified Hausdorff Distance (MHD), and the F-score at a distance threshold $\tau$.

**Chamfer Distance.** The Chamfer Distance measures the average closest-point distance between two point sets. Given $\mathbf{P} = \{p_i\}_{i=1}^N$ and $\mathbf{G} = \{g_j\}_{j=1}^M$, it is defined as:

$$\text{CD}(\mathbf{P}, \mathbf{G}) = \frac{1}{N} \sum_{p \in \mathbf{P}} \min_{g \in \mathbf{G}} \|p - g\|_2 + \frac{1}{M} \sum_{g \in \mathbf{G}} \min_{p \in \mathbf{P}} \|g - p\|_2. \tag{8}$$

This symmetric form accounts for both point sets equally.

**Modified Hausdorff Distance.** The Modified Hausdorff Distance captures the worst-case average deviation between two point clouds. It is defined as:

$$\text{MHD}(\mathbf{P}, \mathbf{G}) = \max \left( \frac{1}{N} \sum_{p \in \mathbf{P}} \min_{g \in \mathbf{G}} \|p - g\|_2, \frac{1}{M} \sum_{g \in \mathbf{G}} \min_{p \in \mathbf{P}} \|g - p\|_2 \right). \tag{9}$$

Unlike the classical Hausdorff distance, which takes the maximum of all pairwise minima, the modified form uses the mean distance in each direction, making it more robust to outliers.

**F-score.** The F-score measures the harmonic mean of precision and recall based on a distance threshold $\tau$. The precision and recall between two point clouds $\mathbf{P}$ and $\mathbf{G}$ are defined as:

$$\text{precision} = \frac{|\{p \in \mathbf{P} \mid \min_{g \in \mathbf{G}} \|p - g\|_2 < \tau\}|}{|\mathbf{P}|}, \tag{10}$$

$$\text{recall} = \frac{|\{g \in \mathbf{G} \mid \min_{p \in \mathbf{P}} \|g - p\|_2 < \tau\}|}{|\mathbf{G}|}. \tag{11}$$

Then the F-score is computed as:

$$\text{F-score} = \frac{2 \cdot \text{precision} \cdot \text{recall}}{\text{precision} + \text{recall}}. \tag{12}$$

This metric reflects the balance between completeness and accuracy of reconstruction, where a higher F-score indicates better geometric alignment at the specified tolerance. In our evaluation, the threshold $\tau$ is set to be 10 cm, aligning with the voxel size.

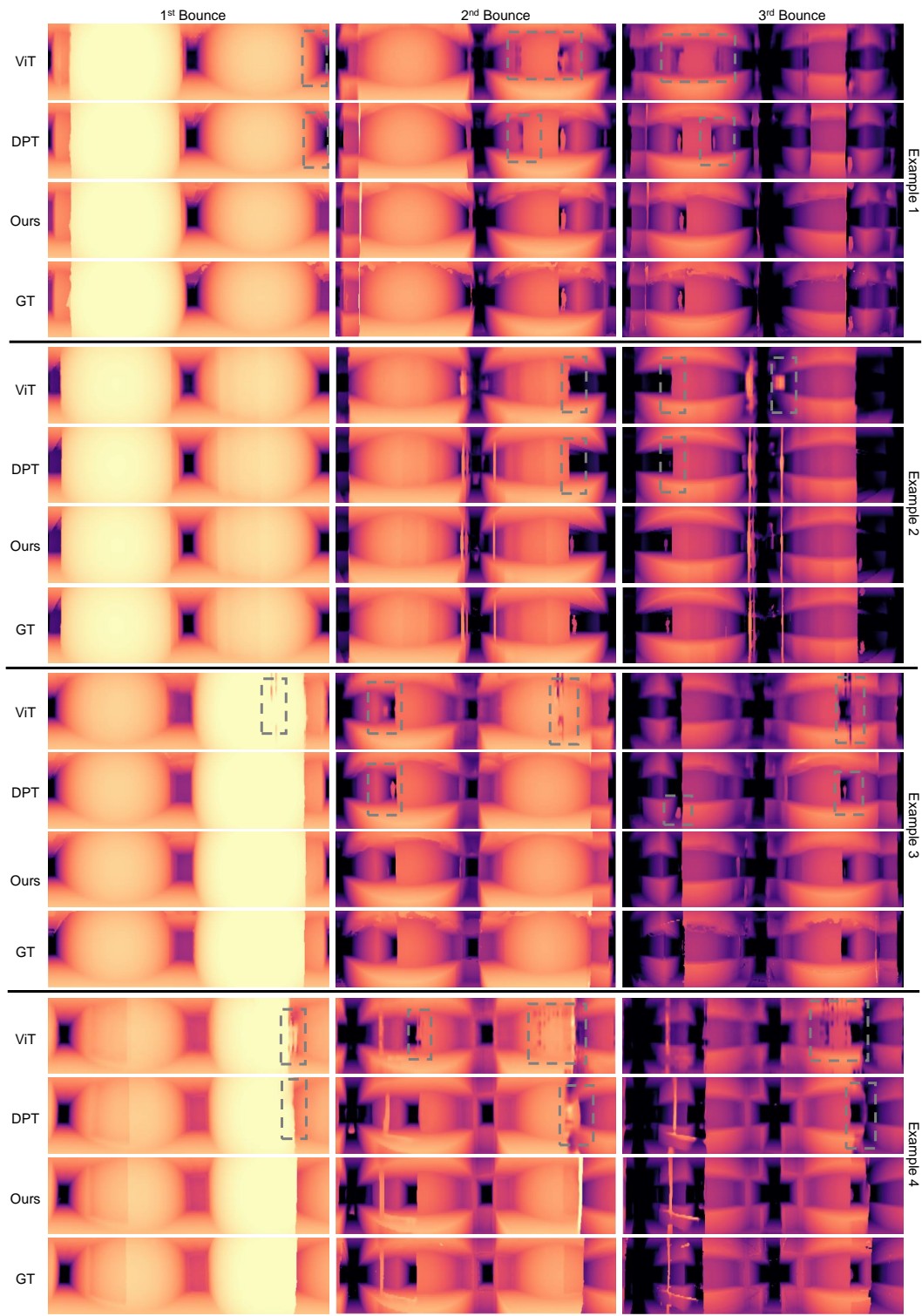

**Figure 10: Multi-return RF imaging results.** We compare our method with baseline approaches for each bounce. Gray boxes highlight artifacts in the baseline predictions.

Together, these three metrics provide a comprehensive evaluation of geometric fidelity between predicted and ground truth point clouds, balancing average distance, structural deviation, and threshold-based overlap.

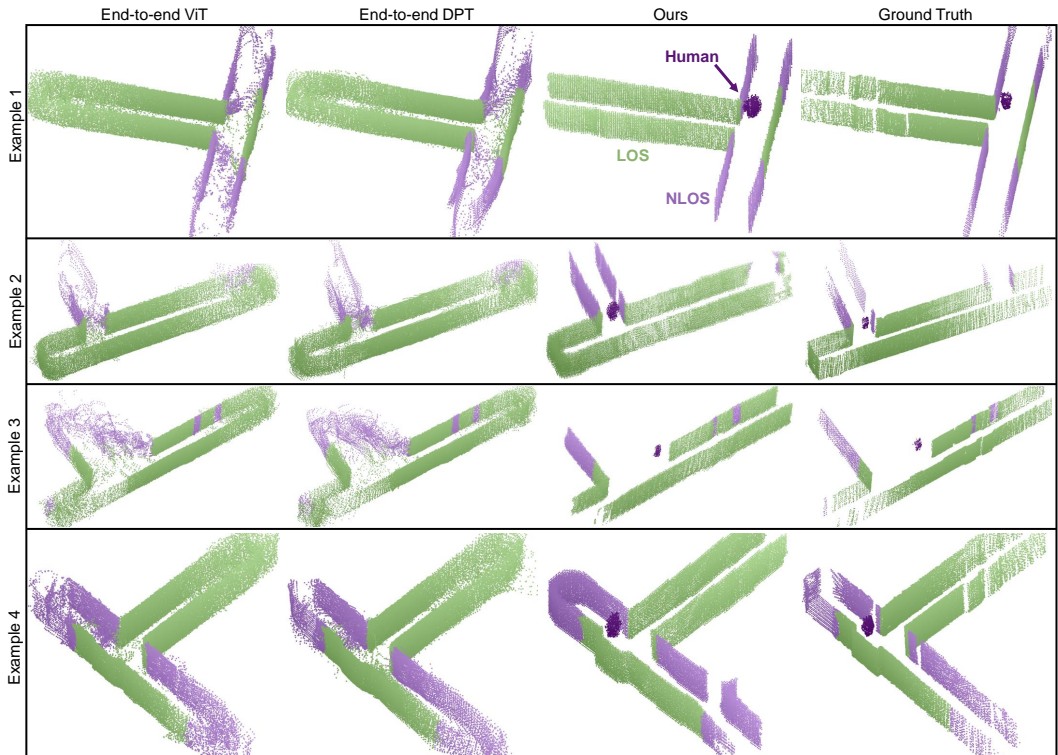

**Figure 11: Scene reconstruction around corners.** We compare our method against two end-to-end transformer-based baselines. Green, light purple, and dark purple points represent LOS geometry, NLOS geometry, and NLOS human, respectively. For all examples, the ceiling and floor are omitted for clearer visualization.

## C  Network Implementation Details

### C.1  Multi-return RF Imaging Model

Our multi-return RF imaging model employs a UNet-based architecture featuring a shared encoder, three separate decoders, and skip connections. Both encoder and decoder modules contain 7 layers built from standard residual blocks [13]. Each encoder layer comprises 4 residual blocks, whereas each decoder layer includes 2 residual blocks. At the end of each encoder layer, feature maps are downsampled by a factor of 2, and correspondingly upsampled at each decoder layer. To maintain sufficient resolution along the elevation dimension, downsampling in this direction is stopped after the fourth layer. Skip connections bridge corresponding layers between the encoder and decoders. Additionally, an initial stem layer using a $7 \times 7$ convolution reduces input channels by a factor of 4. This initial convolution effectively captures the energy leakage pattern along elevation and azimuth from the low-resolution RF heatmap while compressing sparse signals along the range dimension.

The model is trained with a batch size of 8 over 90k iterations using the AdamW optimizer. The optimizer uses parameters $\beta_1 = 0.9$ and $\beta_2 = 0.999$. The learning rate is initiated at $10^{-4}$, with a 1k step warm-up and a cosine annealing schedule. Data augmentation involves random rotations and flips along the azimuth dimension, and per-element Gaussian noise addition to the input heatmap. The heatmaps are log-scaled and normalized to [-1, 1] to mitigate extreme value ranges.

### C.2  Reflection-aware Scene Reconstruction Model

**Ray Tracing.** Our ray-tracing sub-module utilizes hybrid estimation blocks to accurately predict surface normals and reflection directions. There are three types of hybrid estimation blocks: hybrid surface normal estimation block, hybrid surface normal reflection block, and hybrid direction reflection block, as illustrated in Fig. 12. Each hybrid estimation block includes two branches: a physics-guided signal processing branch indicated by the corresponding equation, and a residual feature extraction branch with 4 residual convolution blocks. While Eqn. (2)–(5) describe operations at

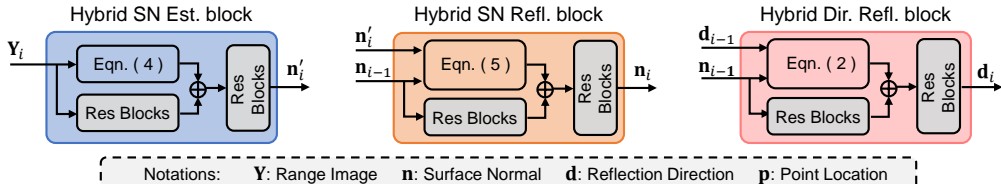

**Figure 12: Structure of the hybrid estimation blocks.** Each of these block contains a physics-guided process indicated by the equation, and a learnable residual feature extraction branch. These feature maps will be concatenated and fused together for the final output (e.g., surface normal or reflection direction).

the level of a single pixel or point, these operations can be applied to the entire image, enabling batch processing that produces surface normals, reflection directions, and point locations as image-like tensors with (x, y, z) as channels. These image-like representations are then processed by the residual feature extraction branch through convolutional layers. The resulting feature maps are concatenated with outputs from the signal processing branch (i.e., Eqn. (2)–(5)) along the channel dimension before being passed to the fusion branch, which consists of 2 residual convolution blocks.

Training is conducted with a batch size of 4 over 60k iterations, using the same optimization strategy as the multi-return RF imaging model. Data augmentation includes random rotations and flips along the azimuth dimension. This sub-module is trained independently and its parameters will be frozen for the subsequent refinement sub-module training.

**Refinement and Reconstruction.** The refinement sub-module features a voxel-based 3D UNet architecture utilizing sparse convolutions, comprising 5 layers each with 2 residual blocks in both encoder and decoder. Spatial dimensions of 20 m×20 m×3.5 m are discretized into 256×256×32 voxels. To ensure sufficient resolution, downsampling along the z (height) dimension stops after reaching 8 voxels at the third layer. Classification and regression heads contain one residual block followed by a 1×1 convolution to adjust output channels. Given the voxel-based nature of this sub-module, the F-score threshold is set to the voxel size (10 cm). Training parameters are identical to those of the ray-tracing sub-module.

### C.3   Baselines

We implement two transformer-based baselines: end-to-end ViT and end-to-end DPT. Both of them use identical training parameters, optimizer settings, and augmentation methods as our multi-return RF imaging model.

**End-to-end ViT.** Our Vision Transformer (ViT [9]) based baseline partitions the RF heatmap into non-overlapping patches of size 4×8 along elevation and azimuth, while retaining all range channels. This operation is similar to the standard technique applied to RGB images. Each patch is flattened and linearly projected into fixed-size tokens. We adopt the ViT-Base architecture in [9] (i.e., 12 layers, 768 hidden size, 3072 MLP size, 12 heads). An output head decodes each token into a 288-dimensional vector reshaped to 9×4×8, representing xyz coordinates across 3 bounces and 4×8 beam directions per patch. For training, it employs per-point L1 loss against ground truth.

**End-to-end DPT.** The Dense Prediction Transformer (DPT [29]) based baseline follows a similar patch embedding and transformer configuration as the ViT baseline. Different from [29], it integrates 6 Reassemble blocks at layers [2, 4, 6, 8, 10, 12]. These blocks adjust feature maps to dimensions $\frac{N_\theta}{s_1} \times \frac{N_\phi}{s_2}$, with $[s_1, s_2]$ sequentially set to [[2,2], [2,4], [4,8], [4,16], [8,32], [8,64]]. Similar to our multi-return RF imaging model, three separate decoders predict point location images for each bounce. For fusion blocks in the decoder, we use feature concatenation instead of direct addition. Per-point L1 loss is used during training.

## D   Social Impacts

The proposed system has significant positive social impacts, primarily enhancing safety and reliability in robotics and autonomous systems. By enabling machines to perceive hidden humans and structures around corners, our approach can substantially reduce accidents involving autonomous vehicles. Additionally, indoor robots, including service robots and emergency-response systems, can operate

more safely and effectively in complex environments around unseen obstacles. However, the increased sensing capability also raises privacy considerations, as the technology could potentially detect individuals without their knowledge or consent. Thus, responsible deployment of our system must include transparency about its usage and careful consideration of privacy protections to balance safety benefits with ethical obligations.

