# OpenReview forum: "Non-Line-of-Sight 3D Reconstruction with Radar"
_NeurIPS.cc/2025/Conference — NeurIPS 2025 poster_

### Official Review · Reviewer_xFjW · 2025-06-30

**Clarity:** 3
**Significance:** 3
**Originality:** 3
**Rating:** 4
**Confidence:** 4

**Summary:**

The paper presents a novel around-the-corner imaging system based on millimeter-wave radar. By employing a series of deep learning models, the system processes radar heatmaps to reconstruct 3D scenes in non-line-of-sight (NLOS) scenarios. The authors collect a custom dataset involving 17 corner configurations and demonstrate that their system can accurately reconstruct scenes, even when objects are occluded from direct view.

**Questions:**

1. Why were vision-based models (e.g., transformers) used as baselines instead of radar-focused multi-bounce imaging methods, which would provide a more meaningful comparison?
2. Could the authors provide detailed information about the diversity of corner scenarios (e.g., widths, angles, distances) in their dataset?
3. Why is the hidden imaging evaluation (human location only) not integrated into the main body of the paper? Would the authors consider moving it and adding clearer evaluation metrics like hit/miss rate?

**Ethical Concerns:**

["NO or VERY MINOR ethics concerns only"]

**Final Justification:**

The rebuttal has addressed most concerns and improved the paper. Although some clarifications on the motivation for full-scene reconstruction and literature context for hidden-human imaging would strengthen it further, I believe the current version shows sufficient novelty and solid evaluation. I am raising my score accordingly and recommend acceptance.

**Limitations:**

yes

**Quality:**

3

**Strengths And Weaknesses:**

Strengths:
- The work is well-motivated, addressing the important problem of NLOS perception.
- The overall system architecture is thoughtfully designed, and the inclusion of a ray tracing module contributes a novel component.
- The dataset collected by the authors adds value and supports meaningful evaluation.

Weaknesses:
- The related work comparison is limited. References to vision transformers justify architectural choices but do not provide a relevant baseline for evaluating the core novelty of the method. It would be more appropriate to compare the method with other radar-based multi-bounce imaging approaches, which are more relevant to the core technical contribution.
- The dataset consists of 17 corners, but additional details about the diversity of these scenes—such as the corridor widths and corner angles—would help assess the generalizability of the results under the proposed experimental setup.
- The evaluation of hidden target imaging (i.e., human location detection without wall depth estimation) should be more prominent. While there is a brief discussion in the supplementary material, this evaluation is crucial and deserves a dedicated section in the main paper, ideally with clear metrics such as hit/miss rate.
- The paper uses “NLOS” broadly, encompassing both behind-wall and around-the-corner scenarios. However, the method focuses specifically on around-the-corner imaging. This distinction should be emphasized early on, potentially starting from the title.

---

> ### Author Rebuttal · Authors · 2025-07-31
>
> We sincerely thank the reviewer for the valuable feedback. In the following, we address each concern with details on our current updates and intended revisions.
> # 1. Missing radar-based multi-bounce imaging baselines
> ### Reviewer’s comments:
> (W1) The related work comparison is limited. References to vision transformers justify architectural choices but do not provide a relevant baseline for evaluating the core novelty of the method. It would be more appropriate to compare the method with other radar-based multi-bounce imaging approaches, which are more relevant to the core technical contribution.
>
> (Q1) Why were vision-based models (e.g., transformers) used as baselines instead of radar-focused multi-bounce imaging methods, which would provide a more meaningful comparison?
> ### Response:
> We thank the reviewer for raising this important point. We agree that radar-based methods would be ideal baselines if appropriate counterparts existed. In fact, we implemented and tested the prior method in [8], which performs NLOS around-the-corner imaging using radar by modeling the relay wall as a specular reflector and applying traditional matched filtering. However, it achieved an NLOS imaging F-score of only around 10%, likely due to the absence of a learning component and lack of scene-level reconstruction capability. For this reason, we did not adopt it as a direct baseline, as the comparison would be both unfair and uninformative.
>
> To further clarify the landscape of related work, we note that existing radar-based NLOS efforts such as [8, A, B] do not support (i) per-pixel 2D range image estimation for NLOS scene geometry, or (ii) dense 3D reconstruction around corners. These two capabilities are central to our contribution and are enabled by our multi-return RF imaging model and reflection-aware reconstruction pipeline.
>
> Given the lack of radar-based methods that align with the scope and output of our system, we chose to compare against well-established vision-based architectures commonly used for range estimation and scene reconstruction. These models serve as strong and representative backbones, allowing us to assess the effectiveness of our architectural choices in isolation.
>
> We agree that future work should aim to benchmark against more radar-based systems as the field matures. In the meantime, our comparisons against general-purpose dense reconstruction models help establish the technical validity of our architectural components, while the qualitative and quantitative results support the novelty of our radar-specific contributions.
>
> ### References:
> - [A] Li, Yuanji, et al. "Millimeter-wave non-line-of-sight imaging." 2022 IEEE 10th Asia-Pacific Conference on Antennas and Propagation (APCAP). IEEE, 2022.
> - [B] Wei, Shunjun, et al. "Nonline-of-sight 3-D imaging using millimeter-wave radar." IEEE Transactions on Geoscience and Remote Sensing 60 (2021): 1-18.
>
> # 2. Details about the diversity of dataset
> ### Reviewer’s comments:
> (W2) The dataset consists of 17 corners, but additional details about the diversity of these scenes—such as the corridor widths and corner angles—would help assess the generalizability of the results under the proposed experimental setup.
>
> (Q2) Could the authors provide detailed information about the diversity of corner scenarios (e.g., widths, angles, distances) in their dataset?
> ### Response:
> We thank the reviewer for highlighting the need for more detailed characterization of our dataset. In response, and as part of our effort to strengthen the experimental validation, we have expanded the dataset from 17 corners across 2 buildings to 32 corners across 5 buildings. This expansion significantly increases the geometric and architectural diversity of the evaluation settings. Detailed information and performance are provided below.
>
> **Corner details:** The five buildings represented in the dataset were constructed between 1906 and 1996, with renovations occurring between 1973 and 2017, capturing a variety of architectural styles and interior layouts. The updated dataset includes 21 T-shaped corners, 5 L-shaped, 5 cross-shaped, and 1 oblique corner with a 45° intersection. Corridor widths span from 1.33 m to 4.63 m, with a mean of 2.16 m and a standard deviation of 0.89 m. This distribution reflects both narrow and wide spaces, contributing to a broad set of scene geometries and occlusion conditions.
>
> **Results:** On the expanded set, both the multi-return RF imaging model and the reflection-aware reconstruction model maintain performance levels comparable to those reported in the original paper, as shown in Tables 1 and 2 below. For the multi-return RF imaging model, the average change in per-pixel range error across the three bounces is within 5 cm of the original dataset. For the reflection‑aware reconstruction, the change in F‑score is 5% relative to the original table. These additions are intended to reduce the risk of site‑specific bias and to document generalization across buildings and layouts.
>
> **Table 1. Performance of the multi-return RF imaging model (mean error, cm↓)**
> | Method | 1st bounce | 2nd bounce | 3rd bounce |
> |:--:|:---:|:---:|:--:|
> |ViT| 9.81|24.30|40.61|
> |DPT| 8.87 |23.00|39.27|
> |Ours| 7.03 |19.03 |31.36|
>
> **Table 2. Results for scene reconstruction**
> |Method|LOS F-score (%)↑|LOS Chamfer (cm)↓|LOS Hausdorff (cm)↓|NLOS F-score (%)↑|NLOS Chamfer (cm)↓|NLOS Hausdorff (cm)↓|
> |:--:|:--:|:--:|:--:|:--:|:--:|:--:|
> |End-to-end ViT|72.4|19.8|13.1|40.4|45.4|28.5|
> |End-to-end DPT|69.8|21.5|14.6|37.0|50.3|31.1|
> |w/o refinement|83.8|13.8|7.3|50.3|43.5|29.3|
> |w/ refinement|85.7|12.2|7.1|54.6 |40.0|28.3|
>
>
> # 3. Evaluation of hidden human imaging
> ### Reviewer’s comments:
> (W3) The evaluation of hidden target imaging (i.e., human location detection without wall depth estimation) should be more prominent. While there is a brief discussion in the supplementary material, this evaluation is crucial and deserves a dedicated section in the main paper, ideally with clear metrics such as hit/miss rate.
>
> (Q3) Why is the hidden imaging evaluation (human location only) not integrated into the main body of the paper? Would the authors consider moving it and adding clearer evaluation metrics like hit/miss rate?
> ### Response:
> We agree with the reviewer that evaluating hidden human imaging is critical and deserves more prominent placement in the main paper. In the original submission, this analysis was included in the supplementary material due to space constraints. However, we now recognize that this evaluation is central to many use cases and plan to integrate it directly into the main body of the revised manuscript.
> In particular, we will present a dedicated evaluation on NLOS human detection and localization, using well-defined and interpretable metrics:
>
> **(1) Detection performance** will be measured using precision, recall, and F1-score (analogous to hit/miss rate), based on whether the predicted human location falls within a 0.5 m threshold of the ground truth.
>
> **(2) Localization accuracy** will be evaluated using the Euclidean distance between predicted and true 3D human positions, reporting both the median and mean error.
>
> For completeness, we also include these results here. Using a 0.5 m detection threshold, our system achieves a precision of 93.8%, recall of 87.9%, and F1-score of 90.7%. The median Euclidean localization error is 13.4 cm, and the mean error is 15.9 cm. These results demonstrate that our system reliably detects and accurately localizes hidden human targets under NLOS conditions.
>
> We appreciate the reviewer’s suggestion, which helps improve the clarity, focus, and completeness of our evaluation.
>
> # 4. Clarifying the scope of “NLOS”
> ### Reviewer’s comments:
> (W4) The paper uses “NLOS” broadly, encompassing both behind-wall and around-the-corner scenarios. However, the method focuses specifically on around-the-corner imaging. This distinction should be emphasized early on, potentially starting from the title.
> ### Response:
> We thank the reviewer for highlighting this important distinction. We agree that our method is specifically designed for around-the-corner scenarios, which represent a particular subset of the broader non-line-of-sight (NLOS) domain. While the term “NLOS” is commonly used in literature to cover a range of occluded perception problems, including through-wall and behind-obstacle settings, our work is focused on the challenges posed by occlusion due to wall geometry at corners. To make this scope more precise and immediately clear to readers, we will make the following revisions:
>
> **(1) Title update:** We will revise the title to explicitly reflect the focus on around-the-corner imaging and reconstruction.
>
> **(2) Abstract and introduction:** We will update both sections to clearly state that the target application is around-the-corner perception, and distinguish it from other forms of NLOS imaging such as through-wall radar.
>
> We believe these clarifications will help set appropriate expectations for readers and more accurately position the contribution of our work. We appreciate the reviewer’s suggestion to improve the precision and clarity of the paper.
>
> # Summary
> We sincerely thank the reviewer for the thoughtful and valuable feedback. These comments have helped strengthen our work in four ways:
> - **(1)** We clarified the novelty of our contributions in the absence of directly comparable radar-based baselines.
> - **(2)** We expanded and better documented the geometric diversity of our dataset.
> - **(3)** We added the hidden human imaging evaluation into the main text with clearer metrics
> - **(4)** We refined the framing of our work to explicitly focus on around-the-corner scenarios within the broader NLOS domain.
>
> We will incorporate all corresponding revisions into the updated manuscript.

---

> > ### Author Response · Authors · 2025-08-08
> > **Kind Follow-up on Rebuttal Discussion**
> >
> > Dear Reviewer, we would like to thank you for your time reviewing our paper. Your comments in the initial review have been very helpful in guiding improvements to the work.
> >
> > $\quad$
> >
> > As the discussion period is nearing its end, we just wanted to kindly follow up to see if you have any thoughts or feedback on our rebuttal. We’d really appreciate hearing them, as your perspective would be valuable as we continue refining the paper.
> >
> > $\quad$
> >
> > Thank you again for your time and contribution.

---

### Official Review · Reviewer_sn4N · 2025-07-02

**Clarity:** 3
**Significance:** 1
**Originality:** 1
**Rating:** 5
**Confidence:** 5

**Summary:**

The authors present an approach to find non-line-of-sight (NLOS) 3D objects in indoor scenes by finding the multi-return range images and then reconstructing the physical scene by providing the reflected signals to their "true" locations. The HoloRadar architecture takes the RF heatmap as input and predcts the scene around corners. They show that they can recover hidden objects in the indoor environment.

**Questions:**

Posted in weakness.

**Ethical Concerns:**

["NO or VERY MINOR ethics concerns only"]

**Final Justification:**

Updated score based on rebuttal.

**Limitations:**

The limitations address the indoor aspect of the problem.

**Quality:**

3

**Strengths And Weaknesses:**

Strengths - The paper is well-written and well-explained.
Weakness -
The authors have missed a paper on exactly the same topic which deals with outdoor environments and predicts trajectory of various objects -
Scheiner, N., Kraus, F., Wei, F., Phan, B., Mannan, F., Appenrodt, N., ... & Heide, F. (2020). Seeing around street corners: Non-line-of-sight detection and tracking in-the-wild using doppler radar. In Proceedings of the IEEE/CVF Conference on Computer Vision and Pattern Recognition (pp. 2068-2077).
In view of this omission, the paper is weakened sufficiently since this deals with the indoor environment only and does not provide any solution for the outdoor clutter.

---

> ### Author Rebuttal · Authors · 2025-07-31
>
> We thank the reviewer for raising a concern on related work, which provides an opportunity for us to clarify our contributions.
> # 1. Clarifying relation to prior work
> ### Reviewer’s comments:
> (W1) The authors have missed a paper on exactly the same topic which deals with outdoor environments and predicts trajectory of various objects - Scheiner, N., Kraus, F., Wei, F., Phan, B., Mannan, F., Appenrodt, N., ... & Heide, F. (2020). Seeing around street corners: Non-line-of-sight detection and tracking in-the-wild using doppler radar. In Proceedings of the IEEE/CVF Conference on Computer Vision and Pattern Recognition (pp. 2068-2077). In view of this omission, the paper is weakened sufficiently since this deals with the indoor environment only and does not provide any solution for the outdoor clutter.
>
> (L1) The limitations address the indoor aspect of the problem.
> ### Response:
> We thank the reviewer for raising this concern and for pointing out the work by Scheiner et al. (CVPR 2020). We appreciate the relevance of this paper to NLOS radar perception, but we respectfully argue that the goals, assumptions, and technical challenges addressed in our work are substantially different. In fact, our contribution builds on and extends beyond the scope of Scheiner et al., tackling a more general and more difficult task: full-scene 3D reconstruction, including static and dynamic elements, in cluttered and multipath-rich indoor environments using only radar.
>
> Below, we summarize the key differences and explain why our method represents a complementary advance.
>
> **Task differences:** Scheiner et al. focus on 2D detection and tracking of moving humans, reporting sparse trajectories on a 2D map. In contrast, our method performs dense 3D reconstruction of both environmental structures and human targets, producing high-resolution range images and full 3D point clouds. This leap in dimensionality and fidelity introduces significant algorithmic and modeling complexity.
>
> **Extra sensor assumptions:** Scheiner et al. rely on a LiDAR sensor to detect and localize the relay wall, a dependency that limits robustness in conditions with visual occlusion (e.g., fog or dust). Our system operates entirely with radar, which is inherently robust to airborne occluders. We estimate all reflective surfaces using only RF signals from a rotating mmWave radar, without relying on auxiliary depth sensors.
>
> **Reflection modeling:** The system in Scheiner et al. handles single-bounce reflections from a known wall, applying rule-based methods to map mirrored points back to the scene. This works in simplified cases but breaks down with multi-bounce propagation. Our model introduces a learned multi-return RF imaging module that produces a distinct 2D image for each bounce, effectively simulating a “multi-return LiDAR”. We then reconstruct the full scene using a deep reflection-aware model that handles complex multi-bounce spatial relationships.
>
> **Environmental complexity:** We respectfully disagree that our focus on indoor environments weakens the significance of our contribution. Indoor scenes also present unique challenges for radar-based perception. Doppler-based detection, which Scheiner et al. rely on, performs well in outdoor environments with fast-moving targets (e.g., pedestrians, cyclists). However, in indoor spaces, people often pause, turn, or remain stationary (e.g., during conversations), resulting in near-zero radial velocity that renders Doppler cues ineffective. Furthermore, the presence of dense clutter and reflective surfaces introduces significant multipath, requiring robust signal modeling. In contrast, outdoor settings are typically sparser, with fewer and weaker multipath interactions.
>
> **Technical significance:** To the best of our knowledge, our system is the first to achieve full-scene 3D reconstruction around corners, capturing both LOS and NLOS structures and humans using only radar. Our two-stage pipeline decouples signal interpretation from spatial reasoning and avoids brittle hand-crafted heuristics. Specifically: (1) Our multi-return RF imaging model resolves multiple bounces and produces high-resolution range images per bounce. (2) Our reflection-aware scene reconstruction model reasons over complex radar point relationships to recover full 3D geometry. Together, these components enable dense reconstruction in environments where prior systems, including Scheiner et al., cannot operate reliably.
>
> # Summary
> While Scheiner et al. make a valuable contribution to radar-based outdoor NLOS detection, their system focuses on 2D tracking of moving targets under simpler signal conditions. Our work targets a more complex and generalizable problem: full-scene 3D reconstruction, including static geometry and dynamic (or static) agents, in multipath-dense, Doppler-sparse indoor environments. Rather than a simplification or omission, our work should be viewed as a complementary extension into a harder domain, and we hope this clarification helps to clearly distinguish the two contributions.

---

> > ### Comment · Reviewer_sn4N · 2025-08-01
> > **Response**
> >
> > **Task differences**
> > Those differences occur mostly because of differences in environments. The 17 indoor scans are pretty similar - thus the learning component is pretty easy and the out-of-distribution (OOD) aspect is a huge problem in this case. The previous paper will not suffer from these drawbacks as its a physics-based approach.
> >
> > **Extra sensor assumptions**
> > The LiDAR is used for outdoor. It would be useful to showcase the power of the proposed approach for an outdoor case to showcase the differences. My assumption is that the noise of mmWave Radar would have an outsized effect and an additional sensor would be required.
> >
> > **Reflection modeling**
> > Again the idea is that the proposed approach uses regular surfaces which are very similar between the training and testing case. In the case of an OOD indoor environment, the proposed approach would suffer.
> >
> > **Technical significance**
> > The proposed approach samples from environments where the surfaces are regular, without any glass surfaces which can have ambiguous reflections or where there is a lot of clutter.

---

> > > ### Author Response · Authors · 2025-08-03
> > >
> > > Dear reviewer, we would like to thank you for the thoughtful follow-up, which helps clarify the key concerns and gives us the opportunity to further substantiate our contributions. From both the initial review and this comment, we identify two core concerns: **(1)** limitations in dataset diversity, and **(2)** the robustness of our method in out-of-distribution (OOD) scenarios. We will address these in turn below.
> > > # 1. Dataset Diversity
> > > To address concerns about environmental homogeneity, we have significantly expanded our dataset from 17 corners across 2 buildings to 32 corners across 5 buildings. The new dataset includes:
> > > - **Corner Types:** 21 T-shaped, 5 L-shaped, 5 cross-shaped, and 1 oblique (45°) corner.
> > > - **Dimensions:** Corner widths ranging from 1.33 m to 4.63 m, with a mean of 2.16 m and a standard deviation of 0.89 m.
> > > - **Architectural Variety:** The buildings span construction dates from 1906 to 1996, with renovations between 1973 and 2017, and differ in architectural style and material composition.
> > >
> > > This expansion directly targets the concern of overfitting to similar, simple environments. Tables 1 and 2 below summarize model performance on this expanded dataset. For the multi-return RF imaging model, the average change in per-pixel range error across the three bounces is within 5 cm of the original dataset. For the reflection‑aware reconstruction, the change in F‑score is 5% relative to the original table. The results remain consistent with our original submission, indicating that our method generalizes well across diverse building geometries and materials.
> > >
> > > **Table 1. Multi-return RF Imaging Accuracy**
> > > |Method|1st bounce error↓| 2nd bounce error↓|3rd bounce error↓|
> > > |:--:|:--:|:--:|:--:|
> > > |ViT|9.81 cm|24.30 cm|40.61 cm|
> > > |DPT|8.87 cm|23.00 cm|39.27 cm|
> > > |Ours|7.03 cm|19.03 cm|31.36 cm|
> > >
> > > **Table 2. Scene Reconstruction Accuracy**
> > > |Method|LOS F-score↑|LOS Chamfer↓|LOS Hausdorff↓|NLOS F-score↑|NLOS Chamfer↓|NLOS Hausdorff↓|
> > > |:--:|:--:|:--:|:--:|:--:|:--:|:--:|
> > > |End-to-end ViT|72.4%|19.8 cm|13.1 cm|40.4%|45.4 cm|28.5 cm|
> > > |End-to-end DPT|69.8%|21.5 cm|14.6 cm|37.0%|50.3 cm|31.1 cm|
> > > |Ours|85.7%|12.2 cm |7.1 cm|54.6%|40.0 cm|28.3 cm|
> > >
> > >
> > > # 2. Method Robustness
> > > The reviewer raised concerns about the robustness of our method in out-of-distribution (OOD) environments, particularly under ambiguous or irregular reflections. To evaluate this, we conducted two experiments aimed at assessing the system’s generalization under challenging conditions.
> > >
> > > ## Experiment 1: Corner Shape Variability
> > >
> > > We first examined performance across different corner shapes, with results summarized in the following Table 3. T-shaped corners yielded the best performance, likely due to their larger reflective surfaces that support more effective multi-bounce signal propagation. Cross-shaped and L-shaped corners, which present more irregular geometries, resulted in slightly reduced performance. However, despite having fewer samples (5 each compared to 21 T-shaped corners), the performance gap remains modest, approximately 10 cm in Chamfer and Hausdorff distances, and around 10% in F-score. This suggests that the system maintains reasonable accuracy even in the presence of structural irregularity.
> > >
> > > **Table 3. Performance on Different Corner Shapes**
> > > | |NLOS Chamfer (cm)↓|NLOS Hausdorff (cm)↓|NLOS F-score (%)↑|
> > > |--|:--:|:--:|:--:|
> > > |T-Shaped|34.1|23.9|61.9|
> > > |Cross-shaped|37.8|24.9|47.8|
> > > |L-shaped|45.3|32.7|52.5|
> > >
> > > ## Experiment 2: Signal Attenuation Analysis
> > > We also evaluated robustness under varying levels of signal attenuation, which reflect differences in reflection behavior due to surface properties such as material composition, roughness, and thickness. Test scenes were grouped by the power drop between the LOS and NLOS paths: low (<10 dB), medium (10–20 dB), and high (>20 dB) attenuation. As expected, performance decreases with greater signal loss. However, the model maintains relatively low geometric errors and strong F-scores even under high attenuation, indicating resilience to difficult reflective conditions.
> > >
> > > **Table 4. Performance Across Attenuation Levels**
> > > |Attenuation|Chamfer (cm)↓|Hausdorff (cm)↓|F-score (%)↑|
> > > |:--:|:--:|:--:|:--:|
> > > |High|39.4|27.1|53.1|
> > > |Medium|37.2|26.4|55.9|
> > > |Low|35.1|25.5|66.1|
> > >
> > > Together, these experiments demonstrate that our method is robust to both geometric variability and signal degradation, supporting its applicability beyond highly regular or idealized conditions.
> > > # Conclusion
> > > We hope this expanded evaluation clarifies that our method does not depend on overly simplistic environments, and that it generalizes well to diverse indoor structures and remains robust under challenging reflective conditions. We appreciate the reviewer’s critique, which has helped strengthen our analysis.

---

> > > > ### Comment · Reviewer_sn4N · 2025-08-05
> > > > **Thanks**
> > > >
> > > > Thanks for the expanded dataset and evaluation. Please try to include these results and the difference with the previous paper. I will increase the score.

---

> > > > > ### Author Response · Authors · 2025-08-05
> > > > > **Thank you reviewer**
> > > > >
> > > > > Dear reviewer, thank you so much for taking the time to review our paper. We really appreciate your support and are glad the expanded evaluation helped clarify the contributions. We'll make sure to include these results and clearly highlight the differences in the final version.

---

### Official Review · Reviewer_sNnL · 2025-07-02

**Clarity:** 3
**Significance:** 3
**Originality:** 3
**Rating:** 5
**Confidence:** 3

**Summary:**

To address the problem that the current system (e.g., LiDAR) and methods are difficult to apply in robots, the authors propose a mmWave radar-based method and system for NLOS 3D reconstruction. The proposed method integrates physics-based modeling and adopts multiple supervised terms to ensure the accuracy of the reconstructions. Experiments show that the proposed method has the capacity to effectively separate signals of each bounce and manage to reconstruct accurate 3D scenario information.

**Questions:**

* Include analyses and results for distinct scenario types. (e.g., outdoor scenarios).
* Include analyses for different material types and light attenuation levels.
* Provide a detailed description of the loss function.
* Include the ablation study referenced in the Weaknesses.

**Ethical Concerns:**

["NO or VERY MINOR ethics concerns only"]

**Final Justification:**

I have decided to raise my score to acknowledge the workload and methodological contributions presented in this paper.

**Limitations:**

See the weaknesses.

**Quality:**

3

**Strengths And Weaknesses:**

### Strengths

* The paper is well constructed.
* The authors adopt self-capturing data to train the network. Although lacking of comparable baseline methods, the authors provides resonable explanations and the results in quantitatively and qualitatively.

### Weaknesses

* The dataset examples provided in the paper seem to solely include indoor scenarios with limited spatial space, which suggests that the proposed method may have restricted practical applicability.
* The paper assumes that the relay wall behaves as a specular reflector. However, light attenuation might also have a relation to occluder material, so the authors should analyze performance under more severe attenuation conditions.
* Concerns regarding the loss function: Are the range images from the three decoders supervised jointly or separately?
* The paper lacks a necessary ablation study. While sparse 3D convolutions reduce the computational cost, the inference time might arise due to the additional cost of computing the rulebook. An alternative method is to downsample features to trade off performance and efficiency. Further theoretical comparisons and explicit FLOPS and parameter counts would further clarify the proposed method’s advantages.

---

> ### Author Rebuttal · Authors · 2025-07-31
>
> We appreciate the reviewer’s thoughtful and constructive feedback. Below, we provide a detailed description of how we have addressed, or intend to address, the points raised.
> # 1. Analyses for distinct scenario types
> ### Reviewer’s comments:
> (W1) The dataset examples provided in the paper seem to solely include indoor scenarios with limited spatial space, which suggests that the proposed method may have restricted practical applicability.
>
> (Q1) Include analyses and results for distinct scenario types. (e.g., outdoor scenarios).
> ### Response:
> **Why we focus on indoor scenarios:** As noted in the limitations section, our current work focuses on indoor around-the-corner scenarios for three main reasons. **(1)** As an initial step toward NLOS 3D scene reconstruction using mmWave radar, indoor environments provide more controlled conditions with less interruptions, enabling more reliable validation of the full pipeline. **(2)** The radar platform we use has limited transmission power. In wide open outdoor environments, signal returns are often sparse or absent across many beam directions, which makes dense reconstruction particularly challenging without longer integration or larger apertures. **(3)** Most existing radar-based dense 3D imaging efforts [8, 18] have focused on indoor scenes. While some outdoor radar works exist, they are either object-specific rather than general-purpose scene-level reconstruction [11], or only focus on 2D detection and tracking [26, A].
>
> **Challenges for indoor scenarios:** While indoor environments are more controlled in terms of external noise, they are far from trivial. They pose unique challenges due to dense multipath effects and clutter from walls, ceilings, floors, and furniture. These complex propagation conditions demand more robust modeling of signal reflections and occlusions. Additionally, human behavior in indoor settings often includes pausing, turning, or standing still. This near-zero radial velocity undermines the effectiveness of Doppler-based detection that is commonly used in outdoor radar detection work [26, A]. These characteristics make robust static and dynamic NLOS perception in indoor scenes especially relevant and nontrivial. For all these reasons, we chose to establish a strong indoor baseline that demonstrates end-to-end feasibility and practical utility for mobile robot navigation in realistic hallway and room environments.
>
> **Additional Experiments for distinct scenario types:** To address the concern about applicability across distinct scenarios, we have expanded our dataset. It now contains 32 corners across 5 buildings, with a total of 28k heatmap scans. The corners span multiple structures, including 21 T‑shaped, 5 L‑shaped, 5 cross-shaped, and 1 oblique (45°) corners. Corner widths range from 1.33 m to 4.63 m, with a mean of 2.16 m and a standard deviation of 0.89 m. The five buildings were constructed between 1906 and 1996, with renovations between 1973 and 2017, and they feature different architectural styles. On the expanded set, both the multi-return RF imaging model and the reflection-aware reconstruction model maintain performance levels comparable to those reported in the original paper, as shown in Tables 1 and 2 below. For the multi-return RF imaging model, the average change in per-pixel range error across the three bounces is within 5 cm of the original dataset. For the reflection‑aware reconstruction, the change in F‑score is 5% relative to the original table. These additions are intended to reduce the risk of site‑specific bias and to document generalization across buildings and layouts.
>
> **Table 1. Performance of the multi-return RF imaging model (mean error, cm↓)**
> |Method|1st bounce|2nd bounce|3rd bounce|
> |:--:|:--:|:--:|:--:|
> |ViT|9.81|24.30|40.61|
> |DPT|8.87|23.00|39.27|
> |Ours|7.03|19.03|31.36|
>
> **Table 2. Results for scene reconstruction**
> |Method|LOS F-score (%)↑|LOS Chamfer (cm)↓|LOS Hausdorff (cm)↓|NLOS F-score (%)↑|NLOS Chamfer (cm)↓|NLOS Hausdorff (cm)↓|
> |:--:|:--:|:--:|:--:|:--:|:--:|:--:|
> |End-to-end ViT|72.4|19.8|13.1|40.4|45.4|28.5|
> |End-to-end DPT|69.8|21.5|14.6|37.0|50.3|31.1|
> |w/o refinement|83.8|13.8|7.3|50.3|43.5|29.3|
> |w/ refinement|85.7|12.2|7.1|54.6 |40.0|28.3|
>
> ### References:
> - [A] Scheiner, Nicolas, et al. "Seeing around street corners: Non-line-of-sight detection and tracking in-the-wild using doppler radar." CVPR. 2020.
>
> # 2. Analyses on relay walls of different attenuation levels
> ### Reviewer’s comments:
> (W2) The paper assumes that the relay wall behaves as a specular reflector. However, light attenuation might also have a relation to occluder material, so the authors should analyze performance under more severe attenuation conditions.
>
> (Q2) Include analyses for different material types and light attenuation levels.
> ### Response:
> We appreciate the reviewer’s observation regarding attenuation effects. In practice, the level of attenuation from a relay wall is governed by several interdependent factors, including material composition, surface roughness, wall thickness, and angle of incidence. Due to the complexity and interaction among these properties, it is difficult to isolate a single variable for controlled experimentation. To address this, we instead evaluate system performance across a range of real-world attenuation levels, measured as the power drop between LOS reflections and the corresponding NLOS signal.
>
> To quantify attenuation, we sample the radar heatmap on the LOS region and subtract the energy observed in the hidden target region. We categorize our test data into three bands based on the attenuation value: low attenuation (<10 dB), medium attenuation (10 to 20 dB), and high attenuation (>20 dB). Table 3 summarizes the reconstruction performance across these categories. As expected, the system performs best in low-attenuation scenarios. However, even under substantial signal loss, it continues to yield low Chamfer and Hausdorff distances and maintains a high F-score. These results demonstrate that our method remains robust and effective even with significant signal attenuation.
>
> **Table 3. Reconstruction performance under different attenuation levels**
> |Attenuation|Chamfer (cm)↓|Hausdorff (cm)↓|F-score (%)↑|
> |:--:|:--:|:--:|:--:|
> |High|39.4|27.1|53.1|
> |Medium|37.2|26.4|55.9|
> |Low|35.1|25.5|66.1|
>
> # 3. Supervision of decoders and loss formulation
> ### Reviewer’s comments:
> (W3) Concerns regarding the loss function: Are the range images from the three decoders supervised jointly or separately?
>
> (Q3) Provide a detailed description of the loss function.
> ### Response:
> We thank the reviewer for pointing this out and apologize for the lack of clarity. The three decoder branches in our multi-return imaging model are supervised jointly. Each branch predicts a range image corresponding to one bounce and is trained with an L1 per-pixel range estimation loss. As shown in Equation (1), the total loss is computed as the sum of the individual L1 losses across the three branches, which reflects joint supervision. We will clarify this in the revised manuscript to avoid confusion.
>
> # 4. Ablation study on 3D conv design and efficiency
> ### Reviewer’s comments:
> (W4) The paper lacks a necessary ablation study. While sparse 3D convolutions reduce the computational cost, the inference time might arise due to the additional cost of computing the rulebook. An alternative method is to downsample features to trade off performance and efficiency. Further theoretical comparisons and explicit FLOPS and parameter counts would further clarify the proposed method’s advantages.
>
> (Q4) Include the ablation study referenced in the Weaknesses.
> ### Response:
> We appreciate the reviewer highlighting this missing analysis. We have conducted a new ablation study to evaluate the trade-offs between performance and efficiency across different model choices. Specifically, we compare four alternatives: (1) our sparse 3D conv model, (2) a full-resolution dense 3D conv model, (3) a 2x and (4) a 4x downsampled dense 3D conv model.
>
> For each variant, we report inference time (on an NVIDIA RTX4070 GPU), FLOPS, parameter count and our metrics. Our sparse 3D convolution model achieves faster inference time compared to the full-resolution dense convolution, with accuracy close to it. Although sparse convolution requires an additional rulebook, which can potentially increase inference time, our scene is highly sparse, allowing sparse convolution to reduce the overall inference time. This validates our choice to balance between computational efficiency and reconstruction quality. With sparse convolution, the end-to-end inference time is 298 ms (38.4 ms for multi-return imaging and 80ms for ray-tracing), which corresponds to 3.35 Hz. Given that the radar rotation cycle is 2 Hz, our system can support real-time operation. These results and comparisons will be included in the revised manuscript.
>
> **Table 4. Runtime and performance for model design choices**
> |Design choices|Inference time (ms)|GFLOPS|#Parameters|NLOS Chamfer (cm)↓|NLOS F-score (%)↑|
> |:--:|:--:|:--:|:--:|:--:|:--:|
> |Full-resolution dense 3D conv|274|622.24|54.7 M|37.5|53.7|
> |2x downsampled 3D conv|121|322.33|54.7 M|37.6|47.1|
> |4x downsampled 3D conv|40|247.61|54.7 M|42.7|29.9|
> |Sparse 3D conv (ours)|180| 225.05|53.1 M|40.0|54.6|
>
> # Summary
> We appreciate the reviewer’s detailed feedback. These comments have helped us improve the paper in four meaningful ways:
> - **(1)** We expanded and diversified the dataset to better assess generalization across different layouts and scene types.
> - **(2)** We conducted new analyses to evaluate performance under varying attenuation levels.
> - **(3)** We clarified the supervision strategy and formulation of the loss function.
> - **(4)** We added an ablation study that compares different 3D conv designs with inference time and performance.
>
> We will integrate these results, tables, and clarifications into the revised manuscript.

---

> > ### Comment · Reviewer_sNnL · 2025-08-04
> > **Thanks for the response**
> >
> > I appreciate the authors’ responses, which resolved the majority of my concerns. I am mainly interested in the discussion of learning-based techniques. Consequently, I have decided to raise my score to acknowledge the workload and methodological contributions presented in this paper.

---

> > > ### Author Response · Authors · 2025-08-04
> > >
> > > We sincerely thank the reviewer for the thoughtful engagement, as well as the time and effort in reviewing our work. We are grateful for the constructive feedback and pleased that the core concerns have been resolved, which has helped improve the clarity and rigor of our paper.

---

### Official Review · Reviewer_rCvz · 2025-07-02

**Clarity:** 4
**Significance:** 3
**Originality:** 2
**Rating:** 5
**Confidence:** 4

**Summary:**

The manuscript presents HoloRadar, a method for 3D reconstruction for both Non-Line-of-Sight (NLOS) and Line-of-Sight (LOS) perceptions, using a rotating mmWave radar sensor. The work consists of a preprocessing step and a two-stage pipeline.

First, data is collected in cycles by rotating a radar to form a cylindrical array covering the whole surroundings. For signals collected after each rotation, beamforming is applied to obtain a 3D heatmap, consisting of Elevations, Azimuths and Range bins.

The first stage of the proposed pipeline uses the 3D heat maps as input. In particular, for each Azimuth-Elevation pair, a deep model estimates a set of bounce distances, each corresponding to a reflection along a multi-return path. Through the use of an encoder/multi-decoder architecture, this multi-return imaging model predicts the range images for each bounce corresponding to the original 3D radar heatmap.

The second stage reconstructs a physical 3D scene through a ray-tracing based learned model an a deep refinement procedure. In particular, the predicted multi-return images are combined with normal estimation at each bounce, to reverse the mirroring effect of bounces. Lastly, residual geometric errors are accounted for by exploiting a voxel-based 3D UNet, which fuses predictions from different bounces and refines the reconstruction with a volumetric representations.

HoloRadar has been deployed on a real-world mobile robot and it has been evaluated across multiple environments, depicting corners, showing promising and accurate results.

**Questions:**

1. [SUGGESTION] According to what discussed in the weaknesses, a more detailed explanation of the experimental settings could refute the possibility of overfitting and non-adaptability to diverse scenes/structures.
    2. [QUESTION] Why did the authors use only the F-Score to assess the reconstruction quality? One could have included Chamfer or Hausdorff distances, or Mesh quality (e.g., watertightness, manifoldness, self-intersections).
    3. [SUGGESTION] The authors should include information about the processing time and requirements of the proposed method.

**Ethical Concerns:**

["NO or VERY MINOR ethics concerns only"]

**Final Justification:**

I have increased my score to acknoledge the work done by the authors in their rebuttal.

**Limitations:**

Yes

**Quality:**

2

**Strengths And Weaknesses:**

Strengths.
    1. The manuscript is well written and detailed, with all the various aspects being fully explained (from the various steps of the processing pipeline, to the description of the training procedure). This clarity allows for replicability (not considering the (un)availability of the dataset).
    2. The presented pipeline is relatively simple, yet effective, both in terms of adopted learning models, training setup (e.g., loss functions) and mathematical formulation.
    3. Both evaluation and ablation study show promising results in terms of metrics.

Weaknesses.
    1. The major concern about this work is related to the dataset gathered and used for the experiments. As far as it can be understood in the manuscript, all corners “captured” belong to the same building / complex, which could introduce a strong learning bias, with possible overfit. A similar discussion can be done on the number of heatmap scans gathered in total (around 1k for each corner), especially if there is no substantial difference in heatmaps associated with the same location (the paper does not give such information).
    2. The manuscript is lacking a more in-depth analysis on the behavior of the proposed method based on the shape (e.g., T-shaped, oblique, rounded), size of corners (e.g., narrow, wide, medium), and number of perceivable entities (e.g., not just people but trash bins, furniture, or general obstacles).
    3. There is no information in the manuscript about the processing requirements of the proposed method, i.e., if it is able to run in real-time (higher or equal frequency of data acquisition rate) and on which hardware.

---

> ### Author Rebuttal · Authors · 2025-07-31
>
> We would like to thank the reviewer for the constructive comments. Below we will detail our responses for what we have done or plan to do addressing the concerns raised.
> ## 1. Dataset diversity and risk of overfitting
> ### Reviewer’s comments:
> (W1) The major concern about this work is related to the dataset gathered and used for the experiments. As far as it can be understood in the manuscript, all corners “captured” belong to the same building / complex, which could introduce a strong learning bias, with possible overfit. A similar discussion can be done on the number of heatmap scans gathered in total (around 1k for each corner), especially if there is no substantial difference in heatmaps associated with the same location (the paper does not give such information).
>
> (Q1) [SUGGESTION] According to what discussed in the weaknesses, a more detailed explanation of the experimental settings could refute the possibility of overfitting and non-adaptability to diverse scenes/structures.
> ### Response:
> We appreciate this concern and have expanded the dataset to make the evaluation more generalizable. The original submission reported 17 corners from 2 buildings, with 6 and 11 corners respectively. We have now extended this to 32 corners across 5 buildings, with building‑wise counts of 6, 11, 5, 5, and 5. This yields a total of 28k heatmap scans. The corner set now includes 21 T‑shaped, 5 L‑shaped, 5 cross‑shaped, and 1 oblique corner at 45°. Corner widths range from 1.33 m to 4.63 m, with a mean of 2.16 m and a standard deviation of 0.89 m. The five buildings were constructed between 1906 and 1996, with renovations between 1973 and 2017, and they feature different architectural styles. These statistics document substantial diversity in geometry and materials. We will add these details to the revised paper.
> To address variation across scans of the same corner, our data were collected with both a moving robot and moving humans, so sensor and target locations vary across acquisitions. At millimeter‑wave frequencies, even small location changes (i.e., millimeter level) can alter signal phase, thus leading to different heatmaps when doing beamforming. We will include a short analysis visualizing scan‑to‑scan heatmap diversity within the same corner in the supplementary material.
>
> **Results.** We summarize our model’s performance on the expanded dataset in the tables below. Both the multi-return RF imaging model and the reflection-aware scene reconstruction model exhibit performance that is consistent with that in the original submission. For the multi-return RF imaging model, the average change in per-pixel range error across the three bounces is within 5 cm of the original dataset. For the reflection-aware reconstruction, the F-score varies by 5%, indicating minimal degradation. These results suggest that our method generalizes robustly across buildings, corner geometries, and scene variations. We will incorporate these updated tables into the revised manuscript.
>
> **Table 1. Performance of the multi-return RF imaging model**
> | Method |1st bounce error↓| 2nd bounce error↓|3rd bounce error↓|
> |:---:|:-----:|:---:|:----:|
> |   ViT  |  9.81 cm |  24.30 cm  | 40.61 cm  |
> |   DPT  | 8.87 cm | 23.00 cm  | 39.27 cm  |
> |  Ours  | 7.03 cm | 19.03 cm  | 31.36 cm  |
>
> **Table 2. Results for scene reconstruction**
> |  Method | LOS F-score↑ | LOS Chamfer↓ | LOS Hausdorff↓ | NLOS F-score↑ | NLOS Chamfer↓ | NLOS Hausdorff ↓|
> |:---:|:----:|:----:|:---:|:----:|:-----:|:--:|
> | End-to-end ViT |  72.4%  | 19.8 cm  |  13.1 cm   |  40.4%| 45.4 cm    |   28.5 cm   |
> | End-to-end DPT |  69.8%  | 21.5 cm   |  14.6 cm |  37.0% |    50.3 cm |  31.1 cm  |
> | w/o refinement |  83.8%  | 13.8 cm  | 7.3 cm  | 50.3%  |  43.5 cm    |  29.3 cm  |
> |  w/ refinement |  85.7%  |    12.2 cm |   7.1 cm  |  54.6% |    40.0 cm    |  28.3 cm |
>
> ## 2. Behavior across corner types, sizes, and perceivable entities
> ### Reviewer’s comments:
> (W2) The manuscript is lacking a more in-depth analysis on the behavior of the proposed method based on the shape (e.g., T-shaped, oblique, rounded), size of corners (e.g., narrow, wide, medium), and number of perceivable entities (e.g., not just people but trash bins, furniture, or general obstacles).
> ### Response:
> **Corner types and sizes:** We agree that an evaluation of system performance on different types of corners is important to show the robustness and generalizability of our method. Therefore, we have conducted additional experiments and summarized the evaluation results on various corners with different shapes (T-shaped, cross-shaped, and L-shaped) and sizes (narrow: < 1.5 m, medium: 1.5–3 m, wide: >3 m) in the following Table 3. The results exhibit modest variation across categories, indicating that our method remains consistently effective regardless of corner layout.
>
> We observe that T-shaped corners yield the best performance, likely due to their larger reflective surfaces, which are more effective for multi-bounce signal propagation. Among the size categories, medium-width corners achieve the lowest reconstruction errors, possibly because narrow corners may introduce stronger multipath interference, while wide corners can reduce signal strength due to increased distance and wider beam dispersion.
>
> **Table 3. Performance on different categories of corners**
> |   |   | NLOS Chamfer (cm)↓|NLOS Hausdorff (cm)↓|NLOS F-score (%)↑|
> |---|---|:---:|:---:|:--:|
> | Shape |   T-Shaped   |  34.1 |  23.9 | 61.9 |
> | | Cross-shaped | 37.8 | 24.9 |  47.8  |
> | |   L-shaped | 45.3 |  32.7 |  52.5 |
> |  Size |Narrow | 39.1 |  27.5 |  57.6 |
> |  | Medium | 33.0 | 21.7 |  57.5 |
> | | Wide | 42.5| 32.8 | 59.2|
>
> **Perceivable entities:** Regarding perceivable entities, our method is formulated as class‑agnostic 3D scene reconstruction around the corner, with humans reported explicitly because (i) they are the primary safety‑critical agents, (ii) their kinematics are dynamic and complex, and (iii) downstream use cases are often human‑centric. All other static entities such as furniture, trash bins, and general obstacles are captured within the reconstructed scene geometry, although we do not assign semantic labels to classify them in the current version. To make this clearer and more informative, we will add:
>
> **(1) Scope clarification.** We will state that our pipeline reconstructs full scene geometry and reports a human category for detection and localization. Static objects appear in the geometry but are not separately classified.
>
> **(2) Qualitative evidence.** A new figure will show the reconstructed geometry, the ground truth geometry, and an RGB image, with the static obstacles highlighted. This figure aims to demonstrate that our system can capture entities other than humans.
>
> **(3) Quantitative evaluation.** To further investigate our system performance on the hidden human target, we have carried out experiments on NLOS human detection and localization. For detection (with 0.5 m threshold), we achieve a precision of 93.8%, recall of 87.9%, and F-score of 90.7%. For localization, the median Euclidean distance error is 13.4 cm and the mean error is 15.9 cm. These results show that the system reliably detects a hidden human and localizes it with good accuracy.
>
> ## 3. Processing requirements and real‑time feasibility
> ### Reviewer’s comments:
> (W3) There is no information in the manuscript about the processing requirements of the proposed method, i.e., if it is able to run in real-time (higher or equal frequency of data acquisition rate) and on which hardware.
>
> (Q3) [SUGGESTION] The authors should include information about the processing time and requirements of the proposed method.
> ### Response:
> We have added a runtime experiment with per‑stage latency measured on an NVIDIA RTX4070 GPU. The multi‑return RF imaging model takes 38.4 ms per heatmap, the learned ray‑tracing model takes 80 ms, and the refinement model takes 180 ms. The end‑to‑end time per scan is 298 ms, corresponding to 3.35 Hz. Since the radar rotation period is 0.5 s per cycle (2 Hz), it shows that our system supports real‑time operation. We will incorporate them in the paper.
>
> ## 4. More metrics beyond F‑score
> ### Reviewer’s comments:
> (Q2) [QUESTION] Why did the authors use only the F-Score to assess the reconstruction quality? One could have included Chamfer or Hausdorff distances, or Mesh quality (e.g., watertightness, manifoldness, self-intersections).
> ### Response:
> We agree that an F‑score alone is not sufficient.  To evaluate our method more thoroughly, we have added new metrics, namely the Chamfer distance and the Hausdorff distance, for both LOS and NLOS evaluations. The results have been summarized in Table 2 and Table 3 (presented above). Our method achieves the best performance across these metrics when compared to ViT and DPT baselines, demonstrating its effectiveness. We will include metric definitions and implementation details in the supplementary material for reproducibility.
>
> ## Summary
> We appreciate the reviewer’s thoughtful feedback. These comments have helped us strengthen the paper in three concrete ways:
> - **(1)** We have expanded and diversified the dataset.
> - **(2)** We have deepened the analysis across corner geometries and entities.
> - **(3)** We have added both runtime measurements and additional reconstruction metrics.
>
> We will integrate these results, tables, and clarifications in the revised manuscript.

---

> > ### Comment · Reviewer_rCvz · 2025-08-06
> > **Comment on the rebuttal**
> >
> > I have read the rebuttal to my comments made by the reviewers and I agree with them the paper would be improved by the additions they propose in the rebuttal. I confirm my positive score for the acceptance of the paper.

---

> > > ### Author Response · Authors · 2025-08-06
> > > **Thank you reviewer**
> > >
> > > Dear reviewer, we would like to thank you for your time in reviewing and discussing the paper. We truly appreciate your support and are glad that the proposed additions helped strengthen the work. We will incorporate them into the revised manuscript.

---

### Note · Authors · 2025-08-12

Dear AC and reviewers:

Thank you for your valuable feedback, which has helped us significantly improve our paper. Below, we summarize each reviewer’s comments and the outcomes of the discussion.

- Reviewer rCvz described our paper as "well written and detailed" and our method as "effective." The reviewer raised questions about dataset diversity, corner shape analysis, and runtime efficiency. In the rebuttal, we collected a more diverse dataset and provided additional analyses. **The reviewer was satisfied with our responses and confirmed a positive score in favor of acceptance.**

- Reviewer sNnL found the paper well constructed and suggested further analyses on corner shapes, wall attenuations, and runtime efficiency. In the rebuttal, we presented results on the expanded dataset with diverse corner shapes and provided further explanation of our findings. These responses addressed the reviewer’s concerns. **The reviewer expressed interest in our learning-based technique and raised the score to acknowledge our contributions.**

- Reviewer sn4N initially noted the omission of a related work. In the rebuttal, we clarified distinctions on multiple aspects such as task scope, environmental complexity, and technical significance. Following the discussion, the reviewer raised additional concerns about dataset diversity and robustness. We responded with results showing consistent performance on the expanded dataset and detailed robustness analyses across varying corner shapes and signal attenuations. **The reviewer recognized these improvements and indicated an increase of the score.**

- Reviewer xFjW considered our work "well-motivated" and addressing "an important NLOS perception problem" with a novel design. In the rebuttal, we clarified baseline methods and added results on the expanded dataset. **Although we did not have a direct discussion, our response to the dataset diversity concern, which was shared by other reviewers, was consistent with and satisfactory to the general feedback.**

### Conclusion
After the rebuttal and discussion phases, **all three participating reviewers supported our work, either maintaining a positive score or raising it.** The expanded dataset and additional robustness experiments directly addressed their suggestions, further strengthening the paper. We sincerely appreciate your time and constructive feedback.

---

### Decision · Program_Chairs · 2025-09-17

**Decision:**

Accept (poster)

**Comment:**

All reviewers unanimously agree that this paper makes significant contributions and recommend acceptance (5,5,5,4). After a careful evaluation, the AC concurs with the reviewers’ assessments and also recommends acceptance.